# UaOR: Uncertainty-aware Observation Reinjection for Vision-Language-Action Models

## Abstract

Vision–Language–Action (VLA) models leverage pretrained Vision–Language Models (VLMs) as backbones to map images and instructions to actions, demonstrating remarkable potential for generalizable robotic manipulation. To improve performance, many methods have been proposed to incorporate additional observation cues (e.g., depth maps, point clouds) and auxiliary modules (e.g., object detectors, encoders), enabling more precise and reliable task execution. Although effective, these approaches often require extensive data collection and additional training or fine-tuning, limiting their flexibility and scalability. Inspired by the finding that Feed-Forward Network (FFN) in language models can act as "key-value memory", we propose **U**ncertainty-**a**ware **O**bservation **R**einjection (**UaOR**), an effective training-free and plug-and-play module for VLA models. Specifically, when the current language model layer exhibits high uncertainty, measured by *Action Entropy*, it reinjects the observation information into the next layer's Feed-Forward Network (FFN) in a blending manner. This mechanism helps VLA models look more clearly on the observation during inference, enabling more confident and faithful action generation. Comprehensive simulation and real-world experiments show that our method consistently improves the performance of heterogeneous VLA models across various tasks and embodiments while incurring minimal computational overhead. Notably, UaOR eliminates the need for extra observation cues or modules, making it a versatile and practical plug-in for existing VLA pipelines.

## 1 Introduction

Recent advancements in Vision–Language Models (VLMs) (Liu et al., 2024; Karamcheti et al., 2024; Beyer et al., 2024; Bai et al., 2025) have delivered remarkable capabilities in multimodal understanding and generalization. Building on these foundations, Vision–Language–Action (VLA) models (Kim et al., 2025b; Black et al., 2024; Kim et al., 2025a; Li et al., 2025b) fine-tuned on large-scale robotic datasets integrate visual observations with language instructions to synthesize low-level control actions, exhibiting strong task execution and robust generalization across diverse robotic manipulation scenarios. Despite these strengths, persistent data-collection bottlenecks and considerable training budgets remain key barriers to scaling and deploying VLA models in practice.

To achieve performance gains, many efforts (Zheng et al., 2024; Bhat et al., 2025; Lin et al., 2025; Dai et al., 2025) have explored interventions at the input level, such as augmenting observations with additional observation priors. TraceVLA (Zheng et al., 2024) introduces visual trace prompting and fine-tunes on 150K robot manipulation trajectories with visual traces. SpatialVLA (Qu et al., 2025) utilizes Ego3D Position Encoding to inject 3D information into the input observations of the visual-language-action model. While effective, such methods often rely on additional observation priors (e.g., visual traces, depth maps), auxiliary modules (e.g., depth/point-cloud encoders) and extensive fine-tuning, rendering them resource-intensive and poorly scalable to larger backbones and datasets. This naturally raises the question: *Is it possible to boost VLA models in a training-free manner, without requiring supplementary observation cues or auxiliary modules?*

To answer this, we begin by recognizing that VLA models inherit strong visual perception and scene understanding from their VLM backbones, which are often underutilized in current designs. Our key intuition is that after ingesting the observation, the model tends to progressively "**forget**" during forward inference. In other words, observation information, comprising visual input and

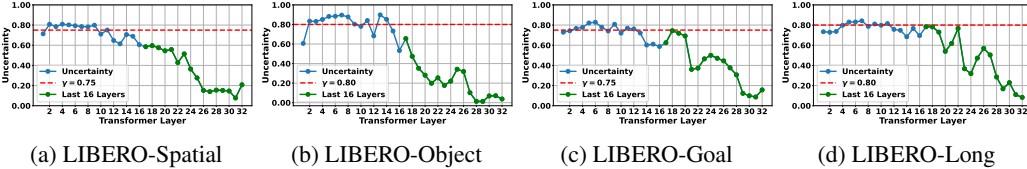

(a) LIBERO-Spatial     (b) LIBERO-Object     (c) LIBERO-Goal     (d) LIBERO-Long

Figure 1: Layer-wise uncertainty of OpenVLA-OFT across four LIBERO task suites. The dashed red line denotes the chosen uncertainty threshold $\gamma$, while the green segment highlights the last 16 layers.

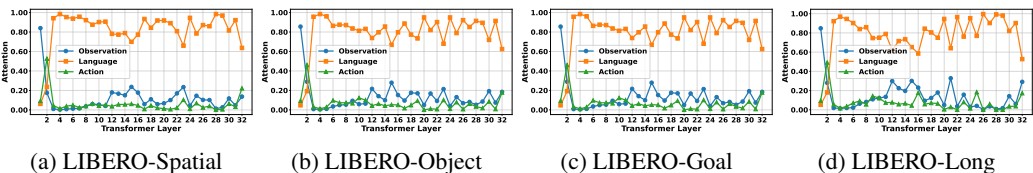

(a) LIBERO-Spatial     (b) LIBERO-Object     (c) LIBERO-Goal     (d) LIBERO-Long

Figure 2: Layer-wise cross-attention from action tokens to observation, language, and action tokens in OpenVLA-OFT across four LIBERO task suites.

proprioceptive state (if available) in our setup, fades as network depth increases, akin to human working-memory decay (Ballard et al., 1995; Horowitz & Wolfe, 1998). This decay may lead to increased uncertainty: Figure 1 reveals an observable rise and sustained high level of uncertainty in the early layers (layers 2-8), a pattern that highly correlates with unfaithful actions (Valle et al., 2025). Consistently, Figure 2 shows that in this uncertainty-rising phase the attention from action tokens to observation tokens drops sharply and then remains at a very low level, indicating that the model rarely consults the observation when predicting actions, and empirically supporting our intuition. Therefore, a natural idea is to reinforce observation information when model exhibits high uncertainty. Inspired by findings that FFNs can act as key-value memory (Geva et al., 2021; Jie et al., 2024; Zou et al., 2024), we adopt the FFN mechanism to extract key features from observation inputs and reinject them into hidden representations, helping the model maintain clear observation throughout inference.

Building on these insights, we propose a lightweight and effective training-free module, **U**ncertainty-**a**ware **O**bservation **R**einjection (UAOR), for VLA models. It computes layer-wise uncertainty via *Action Entropy*, and reinjects observation features into the FFN of the subsequent layer when the uncertainty exceeds a threshold. This blending mechanism reinforces observation information in high-uncertainty regions. Extensive experiments in both simulation and real-world environments show that UAOR consistently improves heterogeneous models across diverse manipulation tasks and embodiments, without retraining or architectural changes. Real-world robotic experiments further validate its practicality and effectiveness. In summary, our main contributions are as follows:

- We introduce *Action Entropy*, a tailored metric to quantify layer-wise uncertainty in VLA models. It reveals a mild rise in uncertainty during the early stages of inference, which we attribute to the model's gradual forgetting of observation information.
- We present **UAOR**, a training-free and plug-and-play module that treats FFN layers as "key-value memory" and reinjects observation features into them when model exhibits high uncertainty, reinforcing the model's attention to observation throughout the inference process.
- We provide rigorous theoretical analysis showing that UAOR enhances the mutual information between hidden states and observation, reduces information bottleneck loss, and lowers expected conditional entropy to mitigate action uncertainty.
- Comprehensive experiments in multiple simulation and real-world environments show that UAOR yields consistent performance gains across various VLA models without relying on extra observation cues or auxiliary modules, while incurring negligible inference overhead.

## 2 RELATED WORK

**Vision-Language-Action Models.** Vision–Language–Action (VLA) models integrate multimodal understanding with action execution, paving the way for more capable robotic systems. A prominent line of works (Brohan et al., 2022; Kim et al., 2025b; Li et al., 2024a; Black et al., 2024) fine-tune

pretrained VLMs on large-scale robot data. RT-2X (Zitkovich et al., 2023) trains a 55B model on the Open X Embodiment (OXE) dataset (Vuong et al., 2023), while OpenVLA (Kim et al., 2025b) fine-tunes a 7B model based on Prismatic (Karamcheti et al., 2024), and $\pi_0$ adapts PaliGemma (Beyer et al., 2024) with a flow matching action head. Another line of works (Ye et al., 2025; Bu et al., 2025; Chen et al., 2025) utilize web-scale videos; e.g., UniVLA (Bu et al., 2025) distills latent actions from internet videos, and EC-Flow (Chen et al., 2025) predicts embodiment-centric flow from unlabeled videos. Recent dual-system architectures (Han et al., 2024; Bu et al., 2024; Bjorck et al., 2025; Cui et al., 2025) separate high-level reasoning (System 2) from low-level control (System 1), showing promise for scalable, general-purpose robotic intelligence.

**Uncertainty in Language Models.** Uncertainty in language models typically reflects the ambiguity and reliability of the predictive distribution. A key indicator is **Entropy** (Ling et al., 2024), where higher values imply lower confidence and potential distribution shift. Farquhar et al. (2024) propose entropy-based uncertainty estimators for LLMs to detect confabulations. Dropout Decoding (Fang et al., 2024) applies uncertainty-guided token dropout principle to input visual tokens for reliability and quality. Recent study of reinforcement learning for LLMs (Wang et al., 2025b) indicates that a minority of high-entropy tokens drives most of the reasoning gains. In the VLA community, there is also a growing focus on uncertainty. Valle et al. (2025) propose Token-Based Entropy (TB-E) as one of the uncertainty metrics for VLA models. Karli et al. (2025) leverages token-level uncertainty to enable uncertainty-aware human intervention during robotic manipulation. In our design, we quantify the uncertainty through action entropy and employ it to evaluate how well the task is executed.

**Visual Augmentation for Manipulation.** Visual augmentation has emerged as a promising strategy to strengthen perception and enhance reliability in robotic control. TraceVLA (Zheng et al., 2024) proposes visual trace prompting to enhance spatial-temporal awareness for generalist robotic policies. PointVLA (Li et al., 2025a) and 3D-CAVLA (Bhat et al., 2025) integrate point clouds and depth maps to improve spatial reasoning capability, respectively. Evo-0(Lin et al., 2025) implicitly injects 3D geometry priors from VGGT (Wang et al., 2025a) into VLA models. AimBot (Dai et al., 2025) overlays shooting lines and scope reticles onto multi-view RGB images to offer auxiliary visual guidance. Compared with these methods, our approach augments observations via the model's inherent FFN layers, without introducing additional visual cues or auxiliary modules.

## 3 METHODOLOGY

### 3.1 PRELIMINARY: REFORMULATION OF FFN

A typical Feed-Forward Network (FFN) in transformer-based models comprises two fully connected layers with an activation in between. Suppose the input hidden states of FFN are $\boldsymbol{h} \in \mathbb{R}^{N \times d}$, where $N$ is the sequence length and $d$ is the hidden dimension, the FFN can be formulated as:

$$\text{FFN}(\boldsymbol{h}) = \phi(\boldsymbol{h}\boldsymbol{W}_1)\boldsymbol{W}_2, \tag{1}$$

where $\phi$ is activation function like ReLU or SiLU, $\boldsymbol{W}_1 \in \mathbb{R}^{d \times D}$ and $\boldsymbol{W}_2 \in \mathbb{R}^{D \times d}$ are the weight matrices of the two FC layers, in usual $D = 4d$. Note that $\boldsymbol{W}_1$ and $\boldsymbol{W}_2$ can be rewritten as:

$$\boldsymbol{W}_1 = (\boldsymbol{k}_1, \boldsymbol{k}_2, ..., \boldsymbol{k}_D), \boldsymbol{W}_2 = (\boldsymbol{v}_1, \boldsymbol{v}_2, ..., \boldsymbol{v}_D)^\top, \tag{2}$$

where $\boldsymbol{k}_i \in \mathbb{R}^d$ and $\boldsymbol{v}_i \in \mathbb{R}^d$ denote entries of key and value, respectively. Then, the FFN can be reformulated as

$$\text{FFN}(\boldsymbol{h}) = \sum_{i=1}^{D} \phi(\langle \boldsymbol{h}, \boldsymbol{k}_i \rangle) \cdot \boldsymbol{v}_i. \tag{3}$$

Therefore, the FFN can be viewed as performing a token-wise key-value lookup mechanism, where each token's hidden state of $\boldsymbol{h}$ serves as the query to calculate its similarity with keys, and gathering values based on the similarity. This formulation closely resembles a key-value memory storing factual knowledge, as suggested in prior work (Geva et al., 2021; Jie et al., 2024; Zou et al., 2024).

### 3.2 PROBLEM FORMULATION

Vision–Language–Action (VLA) models are designed to jointly process observations and language instructions for the purpose of generating appropriate actions for robots. Formally, given the observation $\boldsymbol{o}_t$ at time $t$ and language instruction $\boldsymbol{l}$, a model $\boldsymbol{\pi}$ predicts a temporal action sequence

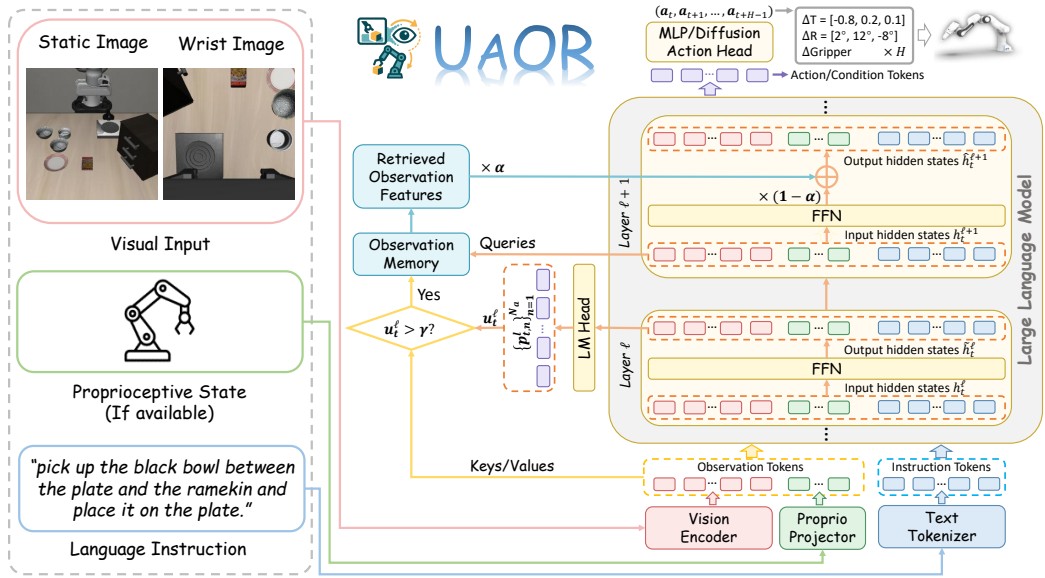

Figure 3: **Overall framework of UAOR**. We compute action entropy at layer $\ell$ to estimate uncertainty. If it exceeds a threshold $\gamma$, we reinject observation features, including visual and proprioceptive features (if available), into the next layer's FFN via a key-value retrieval mechanism, where the input hidden states serve as queries and the observation features act as key-value memory.

$(\boldsymbol{a}_t, \boldsymbol{a}_{t+1}, ..., \boldsymbol{a}_{t+H-1})$ (i.e., action chunk size $H$) for task execution:

$$\pi : (\boldsymbol{o}_t, \boldsymbol{l}) \rightarrow (\boldsymbol{a}_t, \boldsymbol{a}_{t+1}, ..., \boldsymbol{a}_{t+H-1}). \tag{4}$$

In some VLA models (Black et al., 2024; Kim et al., 2025a), the observation $\boldsymbol{o}_t$ includes visual input $\boldsymbol{o}_t^v$ and proprioceptive state $\boldsymbol{o}_t^p$, concatenated as $\boldsymbol{o}_t = [\boldsymbol{o}_t^v; \boldsymbol{o}_t^p]$. In other models, the observation considers only the visual modality, i.e., $\boldsymbol{o}_t = \boldsymbol{o}_t^v$. While in general $\boldsymbol{a}_t$ can represent diverse control schemes and end-effector types, we adopt a simplified setup in this work where actions are defined as 7-DoF vectors corresponding to the gripper's end-effector pose:

$$\boldsymbol{a}_t = [\Delta x, \Delta y, \Delta z, \Delta \phi, \Delta \theta, \Delta \psi, g], \tag{5}$$

where $\Delta x, \Delta y, \Delta z$ are the relative position of the end effector, $\Delta \phi, \Delta \theta, \Delta \psi$ denote the rotation changes, and $g \in \{0, 1\}$ indicates the gripper's open/close state.

### 3.3 UNCERTAINTY-AWARE OBSERVATION REINJECTION

**Uncertainty measured by *Action Entropy*.** Recognizing the central role of entropy as a widely adopted measure of uncertainty, we introduce ***Action Entropy***, a VLA-specific metric that quantifies uncertainty via the entropy of action-related output distributions. Note that current VLA models typically follow two architectures: single-system and dual-system. For single-system models (e.g., OpenVLA-OFT (Kim et al., 2025a)), actions are derived directly from hidden states, either as discrete tokens (256-bin discretization using rare vocabulary tokens) or continuous vectors (via MLP or diffusion heads). We compute entropy directly over the action tokens. For dual-system models (e.g., CogACT (Li et al., 2024a)), System 1 generates actions conditioned on System 2 outputs. We therefore compute entropy over these condition tokens, which guide action generation. Based on this setup, we define layer-wise action entropy at each time step as:

$$\mathcal{H}_{t,n}^{(\ell)} = -\frac{\sum_{i=1}^{K} p_{t,n,i}^{(\ell)} \log p_{t,n,i}^{(\ell)}}{\log K}, \tag{6}$$

where $\boldsymbol{p}_{t,n}^{(\ell)} = \{p_{t,n,i}^{(\ell)}\}_{i=1}^{K}$ denotes the categorical probability distribution over top-$K$ candidate tokens for the $n$-th action or condition token, obtained by projecting the FFN outputs at layer $\ell$ through the language modeling head (LM Head) and normalizing with softmax, which is a standard practice in

---

**Algorithm 1** Uncertainty-aware Observation Reinjection (**UAOR**) for VLA Models

---

**Require:** VLA model $\boldsymbol{\pi}$, observation $\boldsymbol{o}_t$, language instruction $\boldsymbol{l}$
**Output:** Action tokens $\boldsymbol{y}_t$ at time step $t$

1:  **At time step** $t$**:**
2:  **for** $\ell = 1$ to $L - 1$ **do**                          ▷ $L$: total layers
3:       **Compute Uncertainty at Layer** $\ell$**:**
4:       1. Compute action entropy $\mathcal{H}_{t,n}^{(\ell)}$ using FFN output $\tilde{\boldsymbol{h}}_t^{(\ell)}$ at Layer $\ell$     ▷ Eq. 6
5:       2. Compute uncertainty: $u_t^{(\ell)} \leftarrow \frac{1}{N_a} \sum_{n=1}^{N_a} \mathcal{H}_{t,n}^{(\ell)}$           ▷ Eq. 7
6:       **if** $u_t^{(\ell)} > \gamma$ **then**
7:           **Perform Reinjection at Layer** $\ell + 1$**:**
8:           1. Retrieve observation features using $\boldsymbol{h}_t^{(\ell+1)}$: $\text{INJ}_t^{(\ell+1)}(\boldsymbol{o}_t \mid \boldsymbol{h}_t^{(\ell+1)})$     ▷ Eq. 9
9:           2. Blend with original FFN output: $\text{FFN}^{(\ell+1)}(\boldsymbol{h}_t^{(\ell+1)}, \boldsymbol{o}_t)$     ▷ Eq. 8
10:      **end if**
11: **end for**
12: Decode with $\boldsymbol{\pi}(\boldsymbol{o}_t, \boldsymbol{l})$ to obtain $\boldsymbol{y}_t$

---

the "Logit Lens" paradigm (nostalgebraist, 2020; Belrose et al., 2023), For discrete actions, we set $K = 256$ to match the number of action bins, since the model tends to assign higher probability mass to these 256 action tokens. For continuous actions, we likewise fix $K = 256$ for definitional convenience and cross-setting consistency. Based on this formulation, we define the uncertainty of each layer as the average action entropy over all action tokens or condition tokens:

$$u_t^{(\ell)} \;=\; \frac{1}{N_a} \sum_{n=1}^{N_a} \mathcal{H}_{t,n}^{(\ell)}, \tag{7}$$

where $N_a$ is the number of selected tokens (see Appendix B.2 for model-specific settings). Eq. 7 shows higher action entropy indicates greater uncertainty. This formulation enables tracking uncertainty dynamics across layers. Figure 1 visualizes these trends for OpenVLA-OFT across four task suites. We observe a slight increase in uncertainty during the early layers of inference.

**Observation Reinjection with FFN.** As previously discussed, early layers often exhibit high uncertainty. To mitigate this, we introduce **Uncertainty-Aware Observation Reinjection** (**UAOR**), illustrated in Figure 3. Specifically, during the forward pass, we compute the uncertainty $u_t^{(\ell)}$ based on the action entropy at the current layer $\ell$. If this uncertainty exceeds a chosen threshold $\gamma$, it indicates that the model requires clearer observation guidance. Since the forward pass for layer $\ell$ is completed, we perform reinjection at the **subsequent** layer $(\ell + 1)$ to avoid the computational and memory overhead associated with backtracking. Concretely, we treat the encoded observation features as a key-value memory. We use the hidden states entering the FFN at layer $\ell + 1$, denoted as $\boldsymbol{h}_t^{(\ell+1)}$, as queries to attend over this memory. The retrieved features are then blended with the original output of the FFN at layer $\ell + 1$. The formulated process is defined as:

$$\text{FFN}^{(\ell+1)}(\boldsymbol{h}_t^{(\ell+1)}, \boldsymbol{o}_t) = \alpha \text{INJ}_t^{(\ell+1)}(\boldsymbol{o}_t \mid \boldsymbol{h}_t^{(\ell+1)}) + (1 - \alpha)\,\text{FFN}^{(\ell+1)}(\boldsymbol{h}_t^{(\ell+1)}), \tag{8}$$

where $\alpha \in [0, 1]$ is the blending ratio. The retrieved observation features $\text{INJ}_t^{(\ell+1)}$ are computed using $\boldsymbol{h}_t^{(\ell+1)}$ as the queries:

$$\text{INJ}_t^{(\ell+1)}(\boldsymbol{o}_t \mid \boldsymbol{h}_t^{(\ell+1)}) = \sum_{i=1}^{N_o} \phi(\langle \boldsymbol{h}_t^{(\ell+1)}, \boldsymbol{o}_{t,i} \rangle) \cdot \boldsymbol{o}_{t,i}, \tag{9}$$

where $\boldsymbol{o}_t = (\boldsymbol{o}_{t,1}, ..., \boldsymbol{o}_{t,N_o})$ serves as the key-value memory. This design allows the model to dynamically "re-attend" to the observation in the next layer when confusion arises, without needing to halt or backtrack the inference. The complete algorithmic flow is detailed in Algorithm 1.

## 3.4 THEORETICAL ANALYSIS: WHY UAOR WORKS

To understand the effectiveness of UAOR, we provide a theoretical analysis grounded in the following four theorems:

*Notation.* At time step $t$ and layer $\ell + 1$, let $\tilde{\boldsymbol{h}}_t^{(\ell+1)}$ be the vanilla FFN output, $\hat{\boldsymbol{h}}_t^{(\ell+1)}$ the output after applying UAOR (Eq. 8), and $\text{INJ}_t^{(\ell+1)}$ the reinjected observation features (Eq. 9). Let $\boldsymbol{o}_t$ denote the observation, $\boldsymbol{y}_t$ the action tokens, and $\boldsymbol{x}_t$ the full input (observation + language).

**Theorem 3.1** (Observation information gain). *If reinjection is non-degenerate and mixing is near-invertible, then* UAOR *increases the mutual information between the hidden state and observation:*

$$I\left(\hat{\boldsymbol{h}}_t^{(\ell+1)}; \boldsymbol{o}_t\right) \geq I\left(\tilde{\boldsymbol{h}}_t^{(\ell+1)}; \boldsymbol{o}_t\right), \tag{10}$$

*with strict inequality if* $\text{INJ}_t^{(\ell+1)}$ *adds observation-dependent variability.*

**Theorem 3.2** (Action uncertainty reduction). *Assuming a deterministic backbone and stochastic policy head, the conditional entropy over actions is reduced if Theorem 3.1 holds:*

$$H\left(\boldsymbol{y}_t \mid \hat{\boldsymbol{h}}_t^{(\ell+1)}\right) \leq H\left(\boldsymbol{y}_t \mid \tilde{\boldsymbol{h}}_t^{(\ell+1)}\right). \tag{11}$$

**Theorem 3.3** (Information Bottleneck optimization). *Let* $\mathcal{L}(r) = I(r; \boldsymbol{x}_t) - \beta I(r; \boldsymbol{y}_t)$ *be the Information Bottleneck (IB) objective. Then* UAOR *optimizes IB when:*

$$\mathcal{L}(\hat{\boldsymbol{h}}_t^{(\ell+1)}) \leq \mathcal{L}(\tilde{\boldsymbol{h}}_t^{(\ell+1)}) \quad \text{if} \quad \Delta I_{t,y}^{(\ell+1)} \geq \frac{1}{\beta} \Delta I_{t,x}^{(\ell+1)}, \tag{12}$$

*where* $\Delta I_{t,y}^{(\ell+1)} \triangleq I(\hat{\boldsymbol{h}}_t^{(\ell+1)}; \boldsymbol{y}_t) - I(\tilde{\boldsymbol{h}}_t^{(\ell+1)}; \boldsymbol{y}_t), \quad \Delta I_{t,x}^{(\ell+1)} \triangleq I(\hat{\boldsymbol{h}}_t^{(\ell+1)}; \boldsymbol{x}_t) - I(\tilde{\boldsymbol{h}}_t^{(\ell+1)}; \boldsymbol{x}_t).$

**Theorem 3.4** (Benefit of uncertainty-triggered reinjection). *If the entropy-based layer uncertainty* $u_t^{(\ell)}$ *correlates positively with* $H(\boldsymbol{y}_t \mid \tilde{\boldsymbol{h}}_t^{(\ell+1)})$, *then conditioning reinjection on* $u_t^{(\ell)} > \gamma$ *increases the expected relevance of injected information:*

$$\mathbb{E}\left[I\left(\text{INJ}_t^{(\ell+1)}; \boldsymbol{y}_t \mid \tilde{\boldsymbol{h}}_t^{(\ell+1)}\right) \mid u_t^{(\ell)} > \gamma\right] \geq \mathbb{E}\left[I\left(\text{INJ}_t^{(\ell+1)}; \boldsymbol{y}_t \mid \tilde{\boldsymbol{h}}_t^{(\ell+1)}\right)\right]. \tag{13}$$

**Theoretical Integration.** Proofs are provided in Appendix A. These four theorems form a **unified logical framework** explaining why UAOR works: Theorem 3.1 establishes the *mechanism*, guaranteeing that reinjection restores observation information. Theorem 3.2 links this to the *effect*, proving that this information gain mathematically precipitates a reduction in action uncertainty. Theorem 3.3 justifies the *objective* via the Information Bottleneck principle, ensuring that the reinjection contributes valid predictive cues rather than mere noise or redundancy. Finally, Theorem 3.4 validates our *control strategy*, confirming that entropy-based triggering maximizes the expected relevance of the injected information compared to indiscriminate injection. Together, they theoretically ground UAOR as a method that optimizes model confidence through targeted and efficient information restoration.

## 4 EXPERIMENTS

### 4.1 SIMULATION EXPERIMENTS

**Simulation Benchmarks and Baselines.** We conduct evaluations on three widely-used simulation benchmarks in robot learning: LIBERO (Liu et al., 2023), SIMPLER (Li et al., 2025d), and CALVIN (Mees et al., 2022). For these benchmarks, we select several representative VLA models as our baseline: OpenVLA-OFT (7B) (Kim et al., 2025a) and $\pi_0$ (3B) (Black et al., 2024) for LIBERO, CogACT (7B) (Li et al., 2024a) for SIMPLER, and LLaVA-VLA (0.5B) for CALVIN. These baselines differ in both architecture and scale—OpenVLA-OFT and LLaVA-VLA are single-system models, while $\pi_0$ and CogACT follow dual-system design; model sizes range from 0.5B to 7B parameters. This setup enables a comprehensive assessment of UAOR's impact across heterogeneous VLA models, tasks, and embodiments. The main experiments are conducted using three different random seeds to ensure reliability. More implementation details are presented in Appendix B.

**Experimental Results on LIBERO.** Based on OpenVLA-OFT, UAOR delivers consistent gains across all four suites and achieves a remarkable average success rate of **98.0%**, as shown in Table 1. Notably, this performance is comparable to the recent 3D-CAVLA (Bhat et al., 2025) (**98.1%**), yet UAOR eliminates the need for auxiliary depth inputs, CoT reasoning, and fine-tuning, demonstrating superior efficiency. Validating generality, UAOR also consistently boosts the cutting-edge dual-system policy $\pi_0$ (Black et al., 2024) by **+1.5** points on average. The pronounced gains on **LIBERO-Long** across both architectures (**+2.0**) suggest that selectively reinforcing observation information

Table 1: Performance comparison on the LIBERO benchmark. "†" indicates our reproduced results.

| Method | Spatial | Object | Goal | Long | Average |
|---|---|---|---|---|---|
| Octo (fine-tuned) (Ghosh et al., 2024) (*RSS'23*) | 78.9 | 85.7 | 84.6 | 51.1 | 75.1 |
| OpenVLA (Kim et al., 2025b) (*CoRL'24*) | 84.7 | 88.4 | 79.2 | 53.7 | 76.5 |
| TraceVLA (Zheng et al., 2024) (*ICLR'25*) | 84.6 | 85.2 | 75.1 | 54.1 | 74.8 |
| SpatialVLA (Qu et al., 2025) (*RSS'25*) | 88.2 | 89.9 | 78.6 | 55.5 | 78.1 |
| $\pi_0$ + FAST (Pertsch et al., 2025) (*RSS'25*) | 96.4 | 96.8 | 88.6 | 60.2 | 85.5 |
| UniVLA (Bu et al., 2025) (*RSS'25*) | 96.5 | 96.8 | 95.6 | 92.0 | 95.2 |
| CogVLA (Li et al., 2025c) (*NeurIPS'25*) | 98.6 | **98.8** | 96.6 | 95.4 | 97.4 |
| 3D-CAVLA (Bhat et al., 2025) (*arXiv'25*) | 98.2 | **99.8** | 98.2 | 96.1 | 98.1 |
| OpenVLA-OFT† (Kim et al., 2025a) (*RSS'25*) | 98.2±0.4 | 98.2±0.2 | 97.6±0.4 | 94.2±0.2 | 97.1±0.1 |
| *w/* UᴀOR (*Ours*) | **99.0**±0.2 | **98.4**±0.4 | **98.2**±0.4 | **96.2**±0.0 | **98.0**±0.2 |
| Δ | +0.8 | +0.2 | +0.6 | +2.0 | +0.9 |
| $\pi_0$† (Black et al., 2024) (*RSS'25*) | 96.3±0.6 | 96.7±0.7 | 92.9±1.2 | 80.5±1.2 | 91.7±0.5 |
| *w/* UᴀOR (*Ours*) | **97.3**±0.2 | **98.5**±0.2 | **94.3**±0.2 | **82.5**±0.5 | **93.2**±0.1 |
| Δ | +1.0 | +1.8 | +1.4 | +2.0 | +1.5 |

Table 2: Performance comparison on the SIMPLER benchmark. "†" indicates our reproduced results.

| Method | Pick Coke Can | Move Near | Open/Close Drawer | Open and Place | Average |
|---|---|---|---|---|---|
| RT-1 (Brohan et al., 2022) (*arXiv'23*) | 85.7 | 44.2 | 73.0 | 6.5 | 52.4 |
| RT-1-X (Vuong et al., 2023) (*CoRL'23*) | 56.7 | 31.7 | 59.7 | 21.3 | 42.4 |
| RT-2-X (Vuong et al., 2023) (*CoRL'23*) | 78.7 | 77.9 | 25.0 | 3.7 | 46.3 |
| Octo-base (Ghosh et al., 2024) (*RSS'23*) | 17.0 | 4.2 | 22.7 | 0.0 | 11.0 |
| OpenVLA (Kim et al., 2025b) (*CoRL'24*) | 18.0 | 56.3 | 63.0 | 0.0 | 34.3 |
| CogACT† (Li et al., 2024a) (*arXiv'25*) | 92.3±0.3 | 83.7±0.6 | 72.7±0.2 | 43.5±1.0 | 73.1±0.7 |
| *w/* UᴀOR (*Ours*) | **95.0**±0.3 | **87.1**±0.3 | **73.6**±0.4 | **47.2**±0.4 | **75.7**±0.5 |
| Δ | +2.7 | +3.4 | +0.9 | +3.7 | +2.6 |

Table 3: Performance comparison on the CALVIN benchmark. "†" indicates our reproduced results.

| Method | Success Rate (%) | | | | | Avg. Len |
|---|---|---|---|---|---|---|
| | 1/5 | 2/5 | 3/5 | 4/5 | 5/5 | |
| RoboFlamingo (Li et al., 2024b) (*ICLR'24*) | 82.4 | 61.9 | 46.6 | 33.1 | 23.5 | 2.47 |
| GR-1 (Wu et al., 2024) (*ICLR'24*) | 85.4 | 71.2 | 59.6 | 49.7 | 40.1 | 3.06 |
| Vidman Wen et al. (2024) (*NIPS'24*) | 91.5 | 76.4 | 68.2 | 59.2 | 46.7 | 3.42 |
| OpenVLA (Kim et al., 2025b) (*CoRL'24*) | 91.3 | 77.8 | 62.0 | 52.1 | 43.5 | 3.27 |
| VLAS (Zhao et al., 2025a) (*ICLR'25*) | 87.2 | 64.2 | 40.9 | 28.1 | 19.6 | 2.40 |
| LLaVA-VLA† (Zhao et al., 2025b) (*arXiv'25*) | 94.4±0.2 | 82.0±0.8 | 70.8±0.3 | 59.4±0.6 | 48.2±0.4 | 3.55±0.05 |
| *w/* UᴀOR (*Ours*) | **95.5**±0.3 | **84.6**±0.6 | **72.3**±0.5 | **60.7**±0.2 | **49.1**±0.0 | **3.67**±0.03 |
| Δ | +1.1 | +2.6 | +1.5 | +1.3 | +0.9 | +0.12 |

effectively mitigates the "forgetting" of perceptual cues and reduces error accumulation during complex sequential reasoning.

**Experimental Results on SIMPLER.** Table 2 shows that UᴀOR raises the average success rate of CogACT by **+2.6** points (73.1 → 75.7; ∼**3.6%** relative). The improvements are most evident on *Pick coke can* (**+2.7**), *Open top drawer and place apple* (both **+3.7**) and *Move near* (**+3.4**), with a smaller gain on *Open/Close drawer* (+0.9). These tasks demand precise localization and placement under visual clutter, and the results suggest that uncertainty-aware observation reinjection improves scene grounding and decision reliability *without* extra priors or retraining, validating the utility of UᴀOR as a training-free plug-in module .

**Experimental Results on CALVIN.** As demonstrated in Table 3, with LLaVA-VLA on the ABC→D split (Fig. 3), UᴀOR improves success on every track and increases the average consecutive completion length by **+0.12** (3.55 → 3.67; ∼**3.4%** relative). The consistent gains across progressively longer task chains indicate better maintenance of observation fidelity leading to reduced uncertainty in downstream action prediction. Together with LIBERO and SIMPLER, these results substantiate that UᴀOR provides reliable, training-free improvements across heterogeneous VLA architectures,

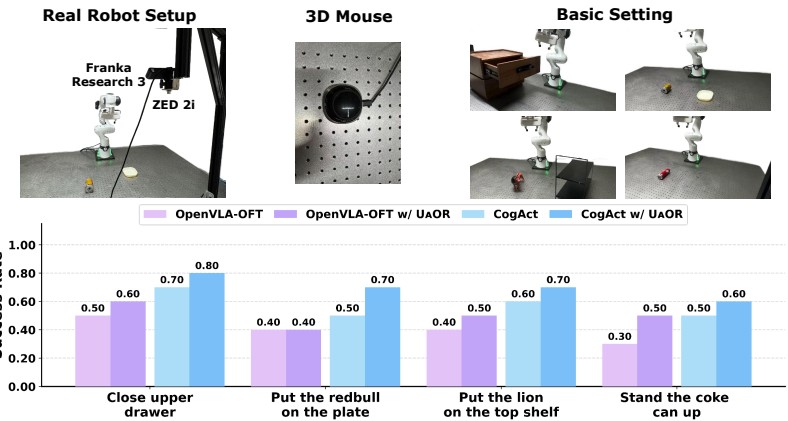

Figure 4: Real-World Setup and Results.

Table 4: Ablation Study on Injection Mechanism, Feature Extraction and Trigger Policy on LIBERO.

| Method / Variant | Feature Extraction | Trigger Policy | Success Rate (%) | | | | |
|---|---|---|---|---|---|---|---|
| | | | Spatial | Object | Goal | Long | Avg. |
| OpenVLA-OFT (Base) | - | - | 98.2 | 98.2 | 97.6 | 94.2 | 97.1 |
| Mean-Residual | Mean Pooling | All Layers | 0.0 | 0.0 | 0.0 | 0.0 | 0.0 |
| Mean-Residual | Mean Pooling | Random | 98.0 | 98.4 | 96.8 | 94.4 | 96.9 |
| Mean-Residual | Mean Pooling | Entropy-based | 0.0 | 0.0 | 0.0 | 0.0 | 0.0 |
| Mean-Blending | Mean Pooling | All Layers | 98.0 | 96.8 | 95.8 | 94.4 | 96.3 |
| Mean-Blending | Mean Pooling | Random | 98.4 | 97.8 | 97.8 | 94.8 | 97.2 |
| Mean-Blending | Mean Pooling | Entropy-based | 98.0 | 97.8 | 97.6 | 93.8 | 96.8 |
| UAOR (All Layers) | Attentive Retrieval | All Layers | 97.8 | 97.6 | 96.2 | 95.2 | 96.7 |
| UAOR (Random) | Attentive Retrieval | Random | 97.8 | 97.6 | 96.4 | 93.6 | 96.4 |
| **UAOR (Ours)** | **Attentive Retrieval** | **Entropy-based** | **99.0** | **98.4** | **98.2** | **96.2** | **98.0** |

tasks, and embodiments. We also provide additional experimental results in Appendix C, including multi-seed evaluations and qualitative visualizations to further show the effectiveness of UAOR.

## 4.2 REAL-WORLD EXPERIMENTS

**Real-World Setup.** We perform real-robot experiments to validate the effectiveness of UAOR in the real world. Our real-robot setup includes a Franka Research 3 robot arm equipped with a parallel-jaw gripper, a static ZED 2i camera, and a 3D mouse (Figure 4). In total, we evaluate on four tasks: 1) *Close the upper drawer*, 2) *Put the redbull on the plate*, 3) *Put the lion on the top shelf*, and 4) *Stand the coke can up*. These tasks range from simple short-horizon placement to complex long-horizon multi-stage manipulation. We fine-tune both OpenVLA-OFT and CogACT on each task using 40 expert trajectories and evaluate each task with 10 test rollouts (see Appendix B.3 for more details).

**Results.** Figure 4 reports the real-world evaluation results on both OpenVLA-OFT and CogACT. For **OpenVLA-OFT**, we observe consistent performance improvements across three of the four tasks, with the average success rate increasing from 40.0% to 50.0% (**+25.0%** relative). The largest gain appears on the most challenging task, *Stand the coke can up* (**+66.7%** relative). Crucially, UAOR demonstrates strong generalizability when applied to **CogACT**. It achieves improvements across *all* four tasks, boosting the average success rate from 57.5% to 70.0% (**+21.7%** relative). Notably, in the *Put the redbull on the plate* task, UAOR increases the success rate by an absolute **20%**. These combined results validate the effectiveness of UAOR in enhancing manipulation robustness and generating faithful actions across different model architectures in real-world scenarios.

## 4.3 ABLATION STUDIES

In this section, we conduct ablation studies on the LIBERO benchmark based on OpenVLA-OFT to investigate the effectiveness of our design choices.

**Ablation on Core Designs.** Table 4 presents a factorial ablation on injection mechanisms, feature extraction, and trigger policies. We define *Mean-Residual* as directly adding mean-pooled observation

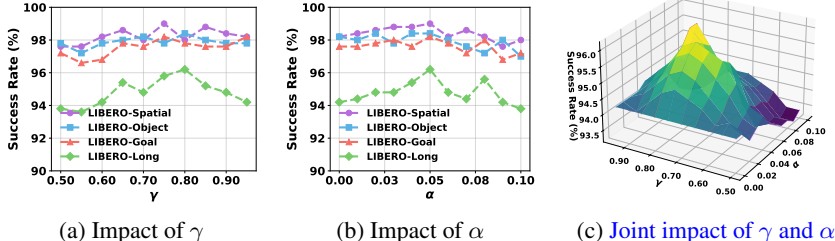

(a) Impact of $\gamma$    (b) Impact of $\alpha$    (c) Joint impact of $\gamma$ and $\alpha$

Figure 5: Impact of uncertainty threshold $\gamma$ and blending factor $\alpha$ across four LIBERO task suites.

features to the FFN's output hidden states and *Mean-Blending* as $\alpha$-blending. Trigger policies include *All Layers*, *Random* (matching the injection rate of Entropy-based), and *Entropy-based* (uncertainty threshold). More details are in Appendix B.4. Results reveal three insights: (1) **Injection Mechanism:** Direct addition causes catastrophic collapse (0.0%) due to severe feature shifts, whereas $\alpha$-blending ensures stability. *Mean-Residual* (Random) survives (96.9%) only because its sparsity allows model recovery in subsequent layers. (2) **Feature Extraction:** Even with stable blending, Mean Pooling (96.3%–97.2%) struggles to surpass the strong baseline (97.1%). This is because averaging assigns equal weight to all observation tokens, failing to distinguish relevant cues. In contrast, UAOR's Attentive Retrieval succeeds by effectively extracting fine-grained context relevant to the current hidden state. (3) **Trigger Policy:** For UAOR, indiscriminate injection (*All Layers/Random*) acts as noise, degrading performance (96.7%/96.4%). Only entropy-based triggering yields improvements (98.0%). Additionally, we have also performed an ablation study in Appendix C.1 to empirically verify the necessity and efficiency of injecting into the next layer's FFN compared to other architectural alternatives. Collectively, these findings validate the effectiveness of the core designs of UAOR.

**Rationale for Action Entropy.** To validate our metric design, we compare UAOR against variants using *Feature Entropy* (hidden state distribution) and a supervised *Learned Head* (linear probe). As shown in Table 5, *Feature Entropy* proves ineffective (96.9%), as it captures representation richness rather than decision uncertainty, often spiking only in the final layers (see Appendix C.2). While the *Learned Head* performs well (97.7%), UAOR achieves superior performance (98.0%) while being en-

Table 5: Comparison of uncertainty metrics on the LIBERO benchmark.

| Metric | Train? | Average |
|---|---|---|
| OpenVLA-OFT | - | 97.1 |
| Feature Entropy | No | 96.9 |
| Learned Head | Yes | 97.7 |
| **Action Entropy** | **No** | **98.0** |

tirely training-free. Additionally, layer-wise probing experiments (Appendix C.2) confirm that intermediate layers already contain significant action semantics (e.g., 78.5% accuracy at Layer 12), validating the use of the frozen LM head as a reliable "rough decoder" for uncertainty estimation.

**Why Select Observation to Reinject?** Table 6 presents an ablation on the type of information reinjected into FFN layers. Results show that reinjecting observation information (i.e., visual and proprioceptive features) yields the most consistent performance improvements. In contrast, reinjecting instruction features—either alone or in combination—leads to no improvement or even performance drops. This suggests that visual and proprioceptive features play a critical role in guiding robot behavior, while also revealing a potential limitation of current VLA models—their insufficient instruction-following capability and tendency to overfit to static language inputs.

Table 6: Reinjection information ablation on LIBERO within OpenVLA-OFT.

| # | Vision | Proprio | Instruction | Average |
|---|---|---|---|---|
| 1 | ✗ | ✗ | ✗ | 97.1 |
| 2 | ✓ | ✗ | ✗ | 97.1 |
| 3 | ✗ | ✓ | ✗ | 96.4 |
| 4 | ✗ | ✗ | ✓ | 96.9 |
| 5 | ✓ | ✓ | ✗ | **98.0** |
| 6 | ✓ | ✗ | ✓ | 96.4 |
| 7 | ✗ | ✓ | ✓ | 97.0 |
| 8 | ✓ | ✓ | ✓ | 96.7 |

**The Impact of $\gamma$ and $\alpha$.** Figure 5 illustrates the effect of varying the uncertainty threshold $\gamma$ and the blending factor $\alpha$ on the performance of OpenVLA-OFT with UAOR. Figures 5a and 5b show the marginal effects when fixing one hyperparameter to its optimal value. To further investigate their interaction, we present a joint sensitivity analysis on LIBERO-Long in Figure 5c. As demonstrated by the 3D surface plot, the performance follows a convex trend, indicating that $\gamma$ and $\alpha$ must be balanced to achieve optimal results. Specifically, we observe two failure modes at the extremes: (1) **Over-correction**: A small $\gamma$ (frequent injection) coupled with a large $\alpha$ (strong mixing) degrades performance, likely by disrupting critical internal representations. (2) **Under-correction**: A large $\gamma$ (rare injection) coupled with a small $\alpha$ (weak mixing) fails to provide sufficient observation guidance.

The distinct peak in Figure 5c confirms our selected parameters lie within the optimal region. In practice, we use an efficient heuristic strategy detailed in Appendix B.2 to determine these values.

## 4.4 MORE ANALYSIS

**Do We Need Token-level Weighting for Visual Inputs?** In UAOR, visual tokens are injected with uniform weights. Inspired by findings that attending to task-relevant regions enhances manipulation (Song et al., 2025), we investigate whether *token-level weighting* improves performance. We design two training-free heuristic weighting schemes that assign each visual token a relevance score based on its similarity to either language instruction tokens ($f_l$) or proprioceptive state tokens ($f_p$):

$w^{(m)} = N_v^{(m)} \cdot \text{softmax}\left(\frac{1}{N_q}\sum_{i=1}^{N_q} f_v^{(m)} f_{q,i}^{\top}\right)$, where

$f_q \in \{f_l, f_p\}$ denotes the query features. Figure 6 visualizes the resulting weights: while language-based weighting focuses on task-relevant objects (Figure 6a), proprio-based weighting exhibits diffuse and semantically ambiguous patterns (Figure 6b), likely due to the lack of explicit alignment between proprioceptive and visual feature spaces. We evaluate these variants on LIBERO with OpenVLA-OFT. As shown in Table 7, neither weighting scheme outperforms the original uniform strategy. Notably, proprio-guided weighting even degrades performance slightly compared to the baseline.

We hypothesize two reasons: 1) Both heuristic schemes lack precise alignment supervision. This is particularly severe for proprioceptive states (e.g., joint angles), which lack intrinsic spatial correspondence with visual tokens, resulting in uninformative weights; 2) VLA models are trained under uniform token weighting and may inherently learn to attend to salient regions, making external heuristic weighting disruptive. In summary, uniform reinjection provides the most robust performance

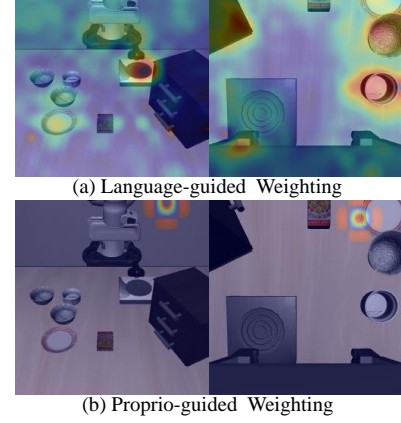

(a) Language-guided Weighting

(b) Proprio-guided Weighting

Figure 6: Visualization of our token-level weighting for visual inputs.

Table 7: Impact of token-level visual weighting based on OpenVLA-OFT.

| Method | Average |
|---|---|
| OpenVLA-OFT | 97.1 |
| *w/* UAOR (Uniform, Default) | **98.0** |
| *w/* UAOR (Language-guided) | 97.3 |
| *w/* UAOR (Proprio-guided) | 96.9 |

**Complexity Analysis.** Although UAOR proves highly effective, an important consideration is its computational cost. We provide a theoretical complexity analysis in Appendix D and evaluate actual runtime overhead through empirical experiments. Specifically, we run 500 rollouts on the LIBERO-Long benchmark using OpenVLA-OFT. As shown in Table 8, applying UAOR results in only a slight throughput drop from 49.7 Hz to 47.3 Hz, and a marginal latency increase from 0.161s to 0.169s. These results indicate that UAOR introduces negligible computational overhead in practice.

Table 8: Comparison of inference overhead between OpenVLA-OFT and OpenVLA-OFT *w/* UAOR. *Throughput* refers to the number of generated actions per second, and *Latency* indicates the inference time per time step.

| Method | Throughput ↑ | Latency ↓ |
|---|---|---|
| OpenVLA-OFT | 49.7 Hz | 0.161 s |
| *w/* UAOR | 47.3 Hz | 0.169 s |

## 5 CONCLUSION

We present UAOR, a lightweight, training-free module designed to boost VLA models. By introducing action entropy as a measure of inference-time uncertainty, UAOR dynamically reinjects observation information into the next-layer FFN when uncertainty is high. This mechanism allows the model to refocus on relevant observation features, leading to more confident and reliable action generation. We provide theoretical analysis demonstrating its efficiency, and validate its effectiveness across a wide range of VLA models, tasks, and embodiments in both simulation and real-world experiments. Without requiring additional observation cues, modules or training, UAOR consistently achieves performance gains with negligible computational overhead, making it a versatile and practical plug-and-play module for existing VLA models.

ETHICS STATEMENT

This work aims to contribute to the advancement of Embodied Intelligence. While our research may have various potential societal implications, none of which we feel must be specifically highlighted here.

REPRODUCIBILITY STATEMENT

We have submitted the relevant code in the supplementary materials. The details of the experimental benchmarks, the baselines, and the simulation and real-world setup can all be found in Section 4.1, Section 4.2, and Appendix B.

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

APPENDIX

## A PROOFS FOR THEORETICAL ANALYSIS: WHY UAOR WORKS

In this section, we provide rigorous proofs for the four theorems stated in Section 3.4, grounded in the Information Bottleneck (IB) theory. We show that UAOR increases the Mutual Information (MI) between hidden states and the observation memory, thereby reducing the conditional entropy given the hidden states, and further improving the Information Bottleneck (IB) objective.

**Preliminaries.** At time step $t$ and layer $\ell+1$, let $\tilde{\boldsymbol{h}}_t^{(\ell+1)}$ be the vanilla FFN output, $\hat{\boldsymbol{h}}_t^{(\ell+1)}$ the output after applying UAOR (Eq. 8), and $\text{INJ}_t^{(\ell+1)}$ the retrieved observation information (Eq. 9). Let $\boldsymbol{o}_t$ be the observation memory, $\boldsymbol{y}_t$ the action tokens, and $\boldsymbol{x}_t$ the full input (observation + language). We write $T_t^{(\ell+1)} \triangleq \big(\tilde{\boldsymbol{h}}_t^{(\ell+1)}, \text{INJ}_t^{(\ell+1)}\big)$ and $\hat{\boldsymbol{h}}_t^{(\ell+1)} = g\big(T_t^{(\ell+1)}\big)$ for the UAOR fusion function induced by Eq. 8. We assume:

- (A1) **Non-degenerate reinjection:** $I\big(\text{INJ}_t^{(\ell+1)}; \boldsymbol{o}_t \mid \tilde{\boldsymbol{h}}_t^{(\ell+1)}\big) > 0$ on a set of non-zero measure.

- (A2) **Near-invertible mixing:** $g$ admits an approximate left inverse $\psi$ with $\mathbb{E}\|\psi(\hat{\boldsymbol{h}}_t^{(\ell+1)}) - T_t^{(\ell+1)}\|_2^2 \le \varepsilon$, implying a strong-DPI type bound (Raginsky, 2016; Polyanskiy & Wu, 2016; Guo et al., 2005)

$$I\big(\hat{\boldsymbol{h}}_t^{(\ell+1)}; Z\big) \ge I\big(T_t^{(\ell+1)}; Z\big) - \kappa_t^{(\ell+1)}(\varepsilon), \quad I\big(\hat{\boldsymbol{h}}_t^{(\ell+1)}; \boldsymbol{y}_t\big) \ge I\big(T_t^{(\ell+1)}; \boldsymbol{y}_t\big) - \eta_t^{(\ell+1)}(\varepsilon), \quad (14)$$

for any $Z$ jointly distributed with $T_t^{(\ell+1)}$, with $\kappa_t^{(\ell+1)}, \eta_t^{(\ell+1)} \to 0$ as $\varepsilon \to 0$ (e.g., Fano/Gaussian bounds).

- (A3) **Target generation:** $\boldsymbol{y}_t$ is generated from $\boldsymbol{x}_t$ through the policy head; conditioned on $\boldsymbol{x}_t$, the rest of the network is deterministic (standard in IB-style analyses (Tishby et al., 2000; Alemi et al., 2017)).

**Lemma A.1** (Layerwise MI decay in the vanilla stack). *For $\ell \ge 1$, $I(\tilde{\boldsymbol{h}}_t^{(\ell+1)}; \boldsymbol{o}_t) \le I(\tilde{\boldsymbol{h}}_t^{(\ell)}; \boldsymbol{o}_t)$.*

*Proof.* Each layer computes $\tilde{\boldsymbol{h}}_t^{(\ell+1)} = f^{(\ell+1)}(\tilde{\boldsymbol{h}}_t^{(\ell)})$ with no direct access to $\boldsymbol{o}_t$, so $\boldsymbol{o}_t \to \tilde{\boldsymbol{h}}_t^{(\ell)} \to \tilde{\boldsymbol{h}}_t^{(\ell+1)}$ is a Markov chain. According to the **Data Processing Inequality (DPI)** (Cover et al., 1991),

if A $\rightarrow$ B $\rightarrow$ C forms a Markov chain, then: $I(A; C) \leq I(A; B)$. Thus we can get $I(\tilde{\boldsymbol{h}}_t^{(\ell+1)}; \boldsymbol{o}_t) \leq I(\tilde{\boldsymbol{h}}_t^{(\ell)}; \boldsymbol{o}_t)$. $\qquad\square$

**Proof of Theorem 3.1** (Observation information gain). At layer $\ell+1$ and time $t$, augment by $T_t^{(\ell+1)} = \big(\tilde{\boldsymbol{h}}_t^{(\ell+1)}, \text{INJ}_t^{(\ell+1)}\big)$. By the chain rule,

$$I\big(T_t^{(\ell+1)}; \boldsymbol{o}_t\big) = I\big(\tilde{\boldsymbol{h}}_t^{(\ell+1)}; \boldsymbol{o}_t\big) + I\big(\text{INJ}_t^{(\ell+1)}; \boldsymbol{o}_t \mid \tilde{\boldsymbol{h}}_t^{(\ell+1)}\big) \geq I\big(\tilde{\boldsymbol{h}}_t^{(\ell+1)}; \boldsymbol{o}_t\big), \qquad (15)$$

with strictness under (A1). Since $\hat{\boldsymbol{h}}_t^{(\ell+1)} = g\big(T_t^{(\ell+1)}\big)$, (A2) yields

$$I\big(\hat{\boldsymbol{h}}_t^{(\ell+1)}; \boldsymbol{o}_t\big) \geq I\big(T_t^{(\ell+1)}; \boldsymbol{o}_t\big) - \kappa_t^{(\ell+1)}(\varepsilon) \geq I\big(\tilde{\boldsymbol{h}}_t^{(\ell+1)}; \boldsymbol{o}_t\big) + I\big(\text{INJ}_t^{(\ell+1)}; \boldsymbol{o}_t \mid \tilde{\boldsymbol{h}}_t^{(\ell+1)}\big) - \kappa_t^{(\ell+1)}(\varepsilon).$$

Letting $\varepsilon \rightarrow 0$ proves $I\big(\hat{\boldsymbol{h}}_t^{(\ell+1)}; \boldsymbol{o}_t\big) \geq I\big(\tilde{\boldsymbol{h}}_t^{(\ell+1)}; \boldsymbol{o}_t\big)$, with strict inequality when $I\big(\text{INJ}_t^{(\ell+1)}; \boldsymbol{o}_t \mid \tilde{\boldsymbol{h}}_t^{(\ell+1)}\big) > 0$. $\qquad\square$

**Proof of Theorem 3.2** (Action uncertainty reduction). Consider the definition of conditional entropy $H(\boldsymbol{y}_t \mid r) = H(\boldsymbol{y}_t) - I(\boldsymbol{y}_t; r)$ (Cover et al., 1991), we have

$$H\big(\boldsymbol{y}_t \mid \hat{\boldsymbol{h}}_t^{(\ell+1)}\big) - H\big(\boldsymbol{y}_t \mid \tilde{\boldsymbol{h}}_t^{(\ell+1)}\big) = -\Big(I\big(\boldsymbol{y}_t; \hat{\boldsymbol{h}}_t^{(\ell+1)}\big) - I\big(\boldsymbol{y}_t; \tilde{\boldsymbol{h}}_t^{(\ell+1)}\big)\Big).$$

By near-invertible mixing (A2) and Eq. 14 with $T_t^{(\ell+1)} = \big(\tilde{\boldsymbol{h}}_t^{(\ell+1)}, \text{INJ}_t^{(\ell+1)}\big)$,

$$I\big(\boldsymbol{y}_t; \hat{\boldsymbol{h}}_t^{(\ell+1)}\big) \geq I\big(\boldsymbol{y}_t; T_t^{(\ell+1)}\big) - \eta_t^{(\ell+1)}(\varepsilon).$$

Applying the chain rule, we get

$$I\big(\boldsymbol{y}_t; T_t^{(\ell+1)}\big) = I\big(\boldsymbol{y}_t; \tilde{\boldsymbol{h}}_t^{(\ell+1)}\big) + I\big(\boldsymbol{y}_t; \text{INJ}_t^{(\ell+1)} \mid \tilde{\boldsymbol{h}}_t^{(\ell+1)}\big).$$

Combining the two displays yields

$$H\big(\boldsymbol{y}_t \mid \hat{\boldsymbol{h}}_t^{(\ell+1)}\big) \leq H\big(\boldsymbol{y}_t \mid \tilde{\boldsymbol{h}}_t^{(\ell+1)}\big) - I\big(\boldsymbol{y}_t; \text{INJ}_t^{(\ell+1)} \mid \tilde{\boldsymbol{h}}_t^{(\ell+1)}\big) + \eta_t^{(\ell+1)}(\varepsilon).$$

Letting $\varepsilon \rightarrow 0$ proves $H\big(\boldsymbol{y}_t \mid \hat{\boldsymbol{h}}_t^{(\ell+1)}\big) \leq H\big(\boldsymbol{y}_t \mid \tilde{\boldsymbol{h}}_t^{(\ell+1)}\big)$, with strict inequality whenever $I\big(\boldsymbol{y}_t; \text{INJ}_t^{(\ell+1)} \mid \tilde{\boldsymbol{h}}_t^{(\ell+1)}\big) > 0$. $\qquad\square$

**Proof of Theorem 3.3** (Information Bottleneck improvement). The Information Bottleneck (IB) objective (Tishby et al., 2000; Alemi et al., 2017) for a representation $r$ is

$$\mathcal{L}(r) = I(r; \boldsymbol{x}_t) - \beta\, I(r; \boldsymbol{y}_t).$$

In particular,

$$\mathcal{L}\big(\tilde{\boldsymbol{h}}_t^{(\ell+1)}\big) = I\big(\tilde{\boldsymbol{h}}_t^{(\ell+1)}; \boldsymbol{x}_t\big) - \beta\, I\big(\tilde{\boldsymbol{h}}_t^{(\ell+1)}; \boldsymbol{y}_t\big), \quad \mathcal{L}\big(\hat{\boldsymbol{h}}_t^{(\ell+1)}\big) = I\big(\hat{\boldsymbol{h}}_t^{(\ell+1)}; \boldsymbol{x}_t\big) - \beta\, I\big(\hat{\boldsymbol{h}}_t^{(\ell+1)}; \boldsymbol{y}_t\big).$$

Let

$$\Delta I_{t,x}^{(\ell+1)} \triangleq I\big(\hat{\boldsymbol{h}}_t^{(\ell+1)}; \boldsymbol{x}_t\big) - I\big(\tilde{\boldsymbol{h}}_t^{(\ell+1)}; \boldsymbol{x}_t\big), \quad \Delta I_{t,y}^{(\ell+1)} \triangleq I\big(\hat{\boldsymbol{h}}_t^{(\ell+1)}; \boldsymbol{y}_t\big) - I\big(\tilde{\boldsymbol{h}}_t^{(\ell+1)}; \boldsymbol{y}_t\big).$$

Then

$$\mathcal{L}\big(\hat{\boldsymbol{h}}_t^{(\ell+1)}\big) - \mathcal{L}\big(\tilde{\boldsymbol{h}}_t^{(\ell+1)}\big) = \Delta I_{t,x}^{(\ell+1)} - \beta\, \Delta I_{t,y}^{(\ell+1)}.$$

Using equation 14 and the chain rule,

$$\Delta I_{t,x}^{(\ell+1)} \leq I\big(T_t^{(\ell+1)}; \boldsymbol{x}_t\big) - I\big(\tilde{\boldsymbol{h}}_t^{(\ell+1)}; \boldsymbol{x}_t\big) + \kappa_t^{(\ell+1)}(\varepsilon) = I\big(\text{INJ}_t^{(\ell+1)}; \boldsymbol{x}_t \mid \tilde{\boldsymbol{h}}_t^{(\ell+1)}\big) + \kappa_t^{(\ell+1)}(\varepsilon),$$

$$\Delta I_{t,y}^{(\ell+1)} \geq I\big(T_t^{(\ell+1)}; \boldsymbol{y}_t\big) - I\big(\tilde{\boldsymbol{h}}_t^{(\ell+1)}; \boldsymbol{y}_t\big) - \eta_t^{(\ell+1)}(\varepsilon) = I\big(\text{INJ}_t^{(\ell+1)}; \boldsymbol{y}_t \mid \tilde{\boldsymbol{h}}_t^{(\ell+1)}\big) - \eta_t^{(\ell+1)}(\varepsilon).$$

Therefore a sufficient condition for $\mathcal{L}\big(\hat{\boldsymbol{h}}_t^{(\ell+1)}\big) \leq \mathcal{L}\big(\tilde{\boldsymbol{h}}_t^{(\ell+1)}\big)$ is

$$\beta\, \Delta I_{t,y}^{(\ell+1)} \geq \Delta I_{t,x}^{(\ell+1)} \quad \Rightarrow \quad \Delta I_{t,y}^{(\ell+1)} \geq \frac{1}{\beta} \Delta I_{t,x}^{(\ell+1)},$$

up to vanishing $\kappa_t^{(\ell+1)}(\varepsilon), \eta_t^{(\ell+1)}(\varepsilon)$ as $\varepsilon \to 0$, which is exactly the criterion stated in Theorem 3.3. Let

$$I_{t,y|\tilde{h}}^{\ell+1,\min} \leq I\big(\mathrm{INJ}_t^{(\ell+1)}; \boldsymbol{y}_t \mid \tilde{\boldsymbol{h}}_t^{(\ell+1)}\big), \qquad I_{t,x|\tilde{h}}^{\ell+1,\max} \geq I\big(\mathrm{INJ}_t^{(\ell+1)}; \boldsymbol{x}_t \mid \tilde{\boldsymbol{h}}_t^{(\ell+1)}\big),$$

be any empirical/theoretical lower and upper bounds, respectively. Then the above inequalities imply

$$\Delta I_{t,y}^{(\ell+1)} \geq I_{t,y|\tilde{h}}^{\ell+1,\min} - \eta_t^{(\ell+1)}(\varepsilon), \qquad \Delta I_{t,x}^{(\ell+1)} \leq I_{t,x|\tilde{h}}^{\ell+1,\max} + \kappa_t^{(\ell+1)}(\varepsilon).$$

Hence a sufficient choice of $\beta$ ensuring $\mathcal{L}(\hat{\boldsymbol{h}}_t^{(\ell+1)}) \leq \mathcal{L}(\tilde{\boldsymbol{h}}_t^{(\ell+1)})$ is

$$\beta \geq \frac{I_{t,x|\tilde{h}}^{\ell+1,\max} + \kappa_t^{(\ell+1)}(\varepsilon)}{I_{t,y|\tilde{h}}^{\ell+1,\min} - \eta_t^{(\ell+1)}(\varepsilon)} \quad \text{provided} \quad I_{t,y|\tilde{h}}^{\ell+1,\min} > \eta_t^{(\ell+1)}(\varepsilon). \tag{16}$$

When $\varepsilon$ is sufficiently small (so that $\kappa_t^{(\ell+1)}(\varepsilon), \eta_t^{(\ell+1)}(\varepsilon) \to 0$), the sufficient condition Eq. 16 simplifies to

$$\beta \geq \frac{I_{t,x|\tilde{h}}^{\ell+1,\max}}{I_{t,y|\tilde{h}}^{\ell+1,\min}} \quad \text{provided} \quad I_{t,y|\tilde{h}}^{\ell+1,\min} > 0.$$

This condition provides a lower bound for $\beta$ to ensure that reinjecting observation information at layer $\ell+1$ reduces the IB objective for VLA models. Satisfying this criterion allows UAOR to effectively optimize the trade-off between compressing task-irrelevant input and retaining observation-relevant information critical for accurate action generation. $\qquad\square$

**Proof of Theorem 3.4** (Benefit of uncertainty-triggered reinjection). Let $u_t^{(\ell)}$ be the entropy-based layer uncertainty; assume it is positively linked to $H\big(\boldsymbol{y}_t \mid \tilde{\boldsymbol{h}}_t^{(\ell+1)}\big)$. Define the *predictive relevance* of the injection at layer $\ell+1$:

$$R_t^{(\ell+1)} \triangleq I\Big(\mathrm{INJ}_t^{(\ell+1)}; \boldsymbol{y}_t \Big| \tilde{\boldsymbol{h}}_t^{(\ell+1)}\Big) \geq 0.$$

Empirically, higher predictive uncertainty correlates with greater expected gains from additional information or computation. Thus, we assume there exists a non-decreasing measurable $\varphi$ such that

$$\mathbb{E}\Big[R_t^{(\ell+1)} \Big| u_t^{(\ell)} = u\Big] = \varphi(u), \qquad \varphi'(u) \geq 0.$$

Then

$$\mathbb{E}\Big[R_t^{(\ell+1)} \Big| u_t^{(\ell)} > \gamma\Big] = \mathbb{E}[\varphi(u) \mid u > \gamma] \geq \mathbb{E}[\varphi(u)] = \mathbb{E}\Big[R_t^{(\ell+1)}\Big],$$

i.e.,

$$\mathbb{E}\Big[I\Big(\mathrm{INJ}_t^{(\ell+1)}; \boldsymbol{y}_t \mid \tilde{\boldsymbol{h}}_t^{(\ell+1)}\Big) \Big| u_t^{(\ell)} > \gamma\Big] \geq \mathbb{E}\Big[I\Big(\mathrm{INJ}_t^{(\ell+1)}; \boldsymbol{y}_t \mid \tilde{\boldsymbol{h}}_t^{(\ell+1)}\Big)\Big].$$

Finally, by the bound proved in Theorem 3.2,

$$H\big(\boldsymbol{y}_t \mid \hat{\boldsymbol{h}}_t^{(\ell+1)}\big) \leq H\big(\boldsymbol{y}_t \mid \tilde{\boldsymbol{h}}_t^{(\ell+1)}\big) - R_t^{(\ell+1)},$$

so triggering on $u_t^{(\ell)} > \gamma$ yields a larger *expected* reduction of $H\big(\boldsymbol{y}_t \mid \hat{\boldsymbol{h}}_t^{(\ell+1)}\big)$ per reinjection call. $\qquad\square$

**Summary.** (A) *Layerwise forgetting* in standard transformer stacks leads to diminishing observation relevance across depth (Lemma A.1). (B) UAOR *recovers* observation dependence at layer $\ell+1$, provably increasing $I(\hat{\boldsymbol{h}}_t^{(\ell+1)}; \boldsymbol{o}_t)$ over the vanilla baseline (Theorem 3.1), which in turn *reduces* conditional entropy $H\big(\boldsymbol{y}_t \mid \hat{\boldsymbol{h}}_t^{(\ell+1)}\big)$ (Theorem 3.2). (C) When the relevance gain $\Delta I_y$ exceeds the scaled compression cost $\frac{1}{\beta}\Delta I_x$, UAOR *lowers* the IB objective, improving the overall informa­tion–efficiency tradeoff (Theorem 3.3). (D) Entropy-based triggering *selectively activates* reinjection in high-uncertainty regions, thereby increasing the expected predictive value of injected content and enhancing per-call entropy reduction (Theorem 3.4).

## B  MORE IMPLEMENTATION DETAILS

### B.1  SIMULATION BENCHMARKS

**LIBERO** (Liu et al., 2023) is a language-conditioned manipulation benchmark that factorizes variation along four axes and evaluates policies under controlled shifts of *geometry*, *object identity*, *goal intent*, and *temporal horizon*. The benchmark provides 4 suites—**Spatial**, **Object**, **Goal**, and **Long**—each containing 10 tasks with 50 human-teleoperated demonstrations per task, yielding a consistent protocol for training and evaluation. These suites focus on distinct reasoning capabilities:

- **LIBERO Spatial** holds objects and goals fixed while perturbing placements and poses, stressing relational language parsing (e.g., left/right, front/behind) and viewpoint robustness.
- **LIBERO-Object** fixes scene layout but varies categories/attributes (type, shape, color), probing category-level generalization and attribute-aware grounding.
- **LIBERO-Goal** keeps geometry and objects constant while changing the intended outcome, testing fine-grained instruction disambiguation and goal-consistent action selection.
- **LIBERO-Long** composes multiple atomic skills into extended procedures across diverse scenes, assessing sequential planning, error recovery, and long-horizon credit assignment.

**SIMPLER** (Li et al., 2025d) is a simulated evaluation suite designed to mirror real-world manipulation with two complementary settings. *Visual Matching (VM)* aligns the simulated scene with its real counterpart (assets, layout, camera), enabling faithful assessment of policies in near-deployment conditions. *Variant Aggregations (VA)* perturbs the VM setup—varying background, lighting, distractors, and table textures—to stress-test robustness and out-of-distribution generalization. For the **Google robot**, both VM and VA include four canonical tasks: 1) *Pick coke can*; 2) *Move near*; and 3) *Open/Close drawer*, and 4) *Open top drawer and place apple*. For the **WidowX robot**, SIMPLER provides the *VM* setting with four tasks: 1) Put spoon on towel, 2) Put carrot on plate, 3) Stack green block on yellow block, and 4) Put eggplant in yellow basket. Evaluation is reported as success rate over standardized rollouts for fair comparison across methods.

**CALVIN** (Mees et al., 2022) is a long-horizon manipulation benchmark built on top of the PyBullet (Coumans & Bai, 2016) simulator and involves a Franka Panda Robot arm that manipulates the scene. It comprises 34 tasks across four environments (A, B, C, and D) and over six hours of teleoperated play data per environment, captured from static and wrist-mounted RGB-D cameras together with tactile signals and proprioception. We adopt the classic and challenging CALVIN ABC→D evaluation protocol, where each model is assessed over 500 rollouts. We report both the overall success rate and the average number of successfully completed sub-tasks (i.e., average length).

### B.2  BASELINES AND SETUP

In this section, we delve into the architectural details of the selected baselines and provide additional information on the experimental setup used throughout our evaluation.

**OpenVLA-OFT** (Kim et al., 2025a) is a high-performance VLA model derived from OpenVLA (Kim et al., 2025b). It incorporates parallel decoding with action chunking, continuous action representation, and an L1 regression objective, leading to substantial improvements in both task performance and inference speed. In our experiments, we use the OpenVLA-OFT variant trained with multimodal inputs consisting of two images (a third-person image and a wrist camera image), the robot's proprioceptive state, and a language instruction. Specifically, the visual and proprioceptive features are concatenated to form the observation features, which are then injected into the Feed-Forward Network (FFN) layers of the language model following our UAOR mechanism. And we compute the action entropy based on all action tokens within the action chunk. We use the hidden states corresponding to the last $N_a = 8 \times 7 = 56$ (action chunk size $H = 8$, action dimension $D_a = 7$) tokens (i.e., positions $[-57 : -1]$) before the final stop token ("") to measure the uncertainty.

$\pi_0$ (Black et al., 2024) employs a flow matching-based architecture built upon the PaliGemma VLM (3B). It processes multimodal inputs (images and language instructions) through the VLM backbone to generate context embeddings (specifically, the Key-Value cache), which then condition a separate action expert for continuous action generation. In our experiments, we inject the visual features into the Feed-Forward Network (FFN) layers of the PaliGemma backbone. Since the flow matching

Table 9: UAOR hyperparameters on simulation and real-world benchmarks

| Benchmark | Base Model | Task / Suite | $\gamma$ | $\alpha$ |
|---|---|---|---|---|
| LIBERO | OpenVLA-OFT | Spatial | 0.75 | 0.05 |
| | | Object | 0.80 | 0.05 |
| | | Goal | 0.75 | 0.05 |
| | | Long | 0.80 | 0.05 |
| | $\pi_0$ | Spatial | 0.20 | 0.05 |
| | | Object | 0.20 | 0.05 |
| | | Goal | 0.20 | 0.05 |
| | | Long | 0.20 | 0.05 |
| SIMPLER | CogACT | Pick coke can | 0.80 | 0.05 |
| | | Move near | 0.80 | 0.05 |
| | | Open/Close drawer | 0.80 | 0.05 |
| | | Open top drawer and place apple | 0.70 | 0.05 |
| CALVIN | LLaVA-VLA | ABC→D | 0.85 | 0.06 |
| Real-World | OpenVLA-OFT | Close upper drawer | 0.75 | 0.05 |
| | | Put the redbull on the plate | 0.80 | 0.05 |
| | | Put the lion on the top shelf | 0.80 | 0.05 |
| | | Stand the coke can up | 0.80 | 0.05 |
| | CogACT | Close upper drawer | 0.80 | 0.05 |
| | | Put the redbull on the plate | 0.80 | 0.05 |
| | | Put the lion on the top shelf | 0.80 | 0.05 |
| | | Stand the coke can up | 0.80 | 0.05 |

head operates in continuous space and does not output discrete action probabilities, we compute the entropy based on the *last token* of the VLM's prefix processing (i.e., position $[-1]$). This metric reflects the backbone's semantic uncertainty regarding the current observation and instruction context before the denoising phase. Consequently, we set $N_a = 1$ in Eq. 7 for this architecture.

**CogACT** (Li et al., 2024a) adopts a componentized dual-system architecture that decouples perception and control. It uses the Prismatic VLM (7B) to extract a cognition token, which conditions a diffusion-based action expert for generating precise actions. CogACT demonstrates state-of-the-art results on the SIMPLER benchmark. In our implementation, since CogACT does not utilize proprioceptive input (i.e., robot joint states), we treat only the visual observation (third-person image) as the modality for observation reinjection. Additionally, we compute the action entropy solely based on the generated cognition token (i.e., positions $[-1]$), which serves as the intermediate representation linking perception and action. Therefore, $N_a = 1$ in Eq. 7 for this setup.

**LLaVA-VLA** (Zhao et al., 2025b) is built on the widely adopted vision-language model LLaVA (Liu et al., 2024), exhibiting stable performance across both simulated and real-world environments. The lightweight variant LLaVA-VLA-0.5b achieves performance comparable to its 7B counterpart based on LLaVA, while incurring significantly lower computational overhead. It incorporates two images (static image and gripper image) and proprioception as input, which we combine as the supplemental observation cues. While LLaVA-VLA adopts action chunking, unlike OpenVLA-OFT, it does not employ parallel decoding and thus generates only one action token per step. Therefore we utilize the last token (i.e., positions $[-1]$, $N_a = 1$) to compute action entropy and uncertainty.

For other baseline methods compared in the main text, we list them for reference and encourage readers to refer to the original papers for further details.

**Hyperparameter Selection Strategy.** We adopt a heuristic strategy to determine the hyperparameters $\gamma$ (uncertainty threshold) and $\alpha$ (blending factor). We begin by analyzing the uncertainty curves (see Figure 1) to obtain a coarse estimate, initially setting $\gamma = 0.80$ for all task suites in LIBERO. Under this preliminary setting, we search for the optimal $\alpha$ and find that $\alpha = 0.05$ yields the best performance across all four LIBERO task suites. Fixing $\alpha$, we then refine $\gamma$ for each individual task by performing a local search around the initial estimate. This progressive narrowing of the search space significantly reduces the tuning overhead while ensuring strong empirical results. We

Table 10: OpenVLA-OFT hyperparameters for real-world fine-tuning.

| Hyperparameter | Value |
|---|---|
| # GPUs | 8 x NVIDIA 4090 (24GB VRAM) |
| learning rate (LR) | 5e-4 |
| total batch size | 8 (1 per GPU) |
| # train steps | 150K |
| input images | 1 third-person camera image |
| input image size | 224 x 224 px |
| use observation history | no (use single-step inputs) |
| LoRA rank | 32 |
| action chunk size | 8 steps (predict 8, execute all 8 open-loop at test time) |
| use proprio (robot state) | yes |
| use FiLM | no |

Table 11: CogACT hyperparameters for real-world fine-tuning.

| Hyperparameter | Value |
|---|---|
| # GPUs | 8 x NVIDIA A100 (80GB VRAM) |
| learning rate (LR) | 2e-5 |
| total batch size | 8 (1 per GPU) |
| input images | 1 third-person camera image |
| input image size | 224 x 224 px |
| VLM backbone | Prism-DinoSigLIP-224px |
| action model type | DiT-B (Diffusion Transformer Base) |
| diffusion steps | 8 (repeated steps) |
| image augmentation | True |
| action chunk size | 16 steps (predict 16, execute all 16 open-loop at test time) |

use the strategy to determine the final hyperparameter settings for both simulation and real-world experiments, as summarized in Table 9.

## B.3 Real-World Setup

Figure 4 illustrates our real-robot setting. The platform comprises a 7-DoF Franka Research 3 robot arm with a parallel-jaw gripper and a ZED 2i stereo camera mounted on a tripod. We collect expert trajectories with a 3D mouse to enable fine-grained and precise manipulation. The four tasks we designed are detailed as follows:

- **Close the upper drawer.** The robot is required to approach the cabinet, locate the upper drawer, and execute a pushing motion to close it fully.
- **Put the redbull on the plate.** The robot needs to identify the Red Bull can, grasp it securely, and place it on the designated plate area with proper orientation.
- **Put the lion on the top shelf.** The robot should pick up the toy lion from the workspace and accurately place it onto the top shelf.
- **Stand the coke can up.** The robot must perform a complex sequence of actions to pick up a horizontally lying cup, reorient it upright, and place it stably on its base.

We fine-tune both OpenVLA-OFT and CogACT on each task using 40 expert trajectories collected with a 3D mouse. The training hyperparameters for OpenVLA-OFT and CogACT are detailed in Table 10 and Table 11, respectively.

## B.4 Ablation on Core Designs

In this section, we provide more details about the ablation study on the core designs of UAOR:

**Mean-Residual:** Directly adds the mean-pooled observation features to the hidden state ($h' = h + o_{mean}$, where $h$ is the original FFN's output hidden states and $o_{mean}$ denotes the mean-pooled

Table 12: Ablation on Injection Timing and Location on LIBERO based on OpenVLA-OFT.

| Injection Timing | Injection Module | Success Rate (%) | | | | | Latency | Overhead |
|---|---|---|---|---|---|---|---|---|
| | | Spatial | Object | Goal | Long | Avg. | | |
| - | *Baseline (No Injection)* | 98.2 | 98.2 | 97.6 | 94.2 | 97.1 | 0.161s | - |
| Current Layer ($\ell$) | Self-Attention (SA) | 98.2 | 98.0 | 97.8 | 95.8 | 97.5 | 0.195s | +21.1% |
| Current Layer ($\ell$) | Feed-Forward (FFN) | 98.6 | 98.2 | 98.0 | 95.8 | 97.7 | 0.182s | +13.0% |
| Next Layer ($\ell + 1$) | Self-Attention (SA) | 98.4 | 98.0 | 97.8 | 94.8 | 97.3 | 0.170s | +5.6% |
| **Next Layer ($\ell + 1$)** | **Feed-Forward (UAOR)** | **99.0** | **98.4** | **98.2** | **96.2** | **98.0** | **0.169s** | **+5.0%** |

observation features) . Represents a naive residual connection. Since the observation tokens and hidden states differ in sequence length, element-wise addition (standard ResNet) is impossible. Therefore, we aggregate observation features via **Mean Pooling** for the residual baselines.

**Mean-Blending:** Blends the mean-pooled observation features using $\alpha$ ($h' = (1 - \alpha)h + \alpha o_{mean}$). Represents a "softer" residual.

**UAOR:** Blends the key observation features relevant to current hidden states via an FFN-like key-value retrieval.

**Trigger Policies: All Layers** injects observation features at every layer of the LLM backbone. **Random** selects a subset of layers uniformly at random for each inference step. To ensure a fair comparison, the number of selected layers matches the average number of layers triggered by the Entropy-based policy (e.g., approximately 30% for LIBERO-Spatial, Object, and Goal, and 20% for LIBERO-Long). **Entropy-based** dynamically triggers injection only at specific layers where the uncertainty measured by action entropy exceeds the threshold $\gamma$, targeting moments of high uncertainty.

# C   MORE EXPERIMENTAL RESULTS

## C.1   ABLATION ON INJECTION TIMING AND LOCATION

To validate the rationale behind our specific design choices—namely, the "one-layer delay" strategy and the selection of the Feed-Forward Network (FFN) as the injection site—we conduct a detailed ablation study comparing different injection timings and module locations on the LIBERO benchmark based on OpenVLA-OFT. The results are summarized in Table 12.

**(1) Why "One-Layer Delay"? (Efficiency & Effectiveness).** We compare injecting into the *Current Layer* ($\ell$) versus our proposed *Next Layer* ($\ell + 1$) strategy.

- **Effectiveness:** As shown in Table 12, injecting into the *Current FFN* (97.7%) and *Next FFN* (98.0%) yields comparable performance. This is because the underlying operation is mathematically identical (using the FFN's input to retrieve observation features and blending them with the original output). The slight edge for *Next Layer* may stem from using more processed hidden states as the queries.
- **Efficiency:** Despite similar success rates, the *Current Layer* strategies incur significantly higher computational overhead. Injecting into the current FFN requires fetching the cached FFN input from memory to perform retrieval, introducing **Memory I/O overhead** and pipeline stalls (0.182s, +13.0%). Injecting into the current Self-Attention (SA) is even costlier (0.195s, +21.1%) as modifying the SA output necessitates a **re-computation** of the subsequent FFN block. In contrast, our *Next Layer* design allows for a seamless "look-ahead" injection without backtracking or re-computation, achieving the optimal efficiency (0.169s, +5.0%).

**(2) Why FFN over Self-Attention?** Comparing *Next Layer FFN* (98.0%) with *Next Layer SA* (97.3%) confirms that the FFN is the superior injection site. We hypothesize the reasons as follows: FFNs structurally function as **Key-Value Memories** (Geva et al., 2021; Jie et al., 2024), making them the natural component for retrieving and storing external information (observation). In contrast, Self-Attention focuses on token-to-token contextualization; injecting external features there may dilute the attention distribution, leading to slightly inferior performance.

## C.2 Detailed Analysis of Uncertainty Metrics

**Layer-wise Probing Verification.** To validate that intermediate hidden states contain meaningful action information, we fine-tune linear lm heads (for discrete actions, the lm head is also the action head) at intermediate layers within the OpenVLA-OFT backbone on the LIBERO-Long suite. As shown in Figure 7, the success rate rises significantly in early-to-mid layers (e.g., reaching 78.5% by Layer 12), confirming that intermediate hidden states already contain significant action-relevant information. This validates our design of using the frozen LM head as a "rough decoder": since the features are semantically aligned with the action space, the resulting entropy serves as a reliable proxy for the model's current uncertainty.

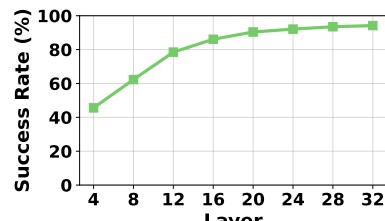

Figure 7: Layer-wise probing results on LIBERO-Long based on OpenVLA-OFT.

**Feature Entropy vs. Action Entropy.** We analyzed the layer-wise trend of Feature Entropy (entropy of the softmax-normalized hidden state vector) on LIBERO-Long based on OpenVLA-OFT. As illustrated in Figure 8, Feature Entropy remains negligible ($\approx 0$) in middle layers and spikes drastically only in the final layers. This trend contradicts the expected behavior of decision uncertainty (which should decrease). Instead, it reflects feature activation richness. Consequently, Feature Entropy fails to trigger reinjection when the model is actually confused, rendering it ineffective compared to our Action Entropy.

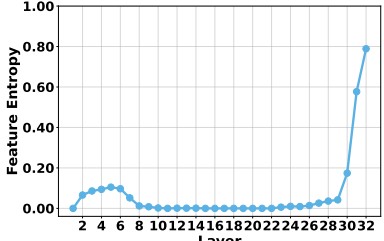

Figure 8: Layer-wise feature entropy on LIBERO-Long.

## C.3 Visualizations of Simulation and Real-World Results

We present additional qualitative results in both simulation and real-world settings to showcase the effectiveness of UAOR. All experiments are conducted within the OpenVLA-OFT framework. As illustrated in Figure 9 and Figure 10, the model successfully completes diverse multi-stage manipulation tasks under varying object configurations and instruction formulations. Benefiting from the uncertainty-aware reinjection mechanism, UAOR helps the model maintain focused attention on key observations during inference, enhancing scene understanding and decision confidence. These visualizations highlight the practicality and adaptability of our method in robotic manipulation.

## D Theoretical Complexity Analysis

For simplicity, we only consider the computational overhead of the Multi-Head Self-Attention (MHSA) and Feed-Forward Network (FFN) blocks in a language model backbone. Let $L$, $N$, and $D$ denote the number of transformer layers, the length of the token sequence, and the hidden dimension, respectively. Following prior works (Jie et al., 2024; Yang et al., 2025), the floating-point operations (FLOPs) for MHSA and FFN in one layer are approximately $8ND^2 + 4N^2D$ and $16ND^2$, respectively. Thus, the total FLOPs of the language model backbone are:

$$\text{FLOPs}_{\text{LM}} \approx L \cdot \left[ (8ND^2 + 4N^2D) + 16ND^2 \right] = L \cdot (24ND^2 + 4N^2D). \tag{17}$$

The additional computational overhead introduced by UAOR consists of two parts: (1) the **projection cost** to compute action entropy, and (2) the **reinjection cost** when uncertainty exceeds the threshold.

**Projection Cost.** To compute the action entropy, we project the hidden states of action-related tokens into the vocabulary space using the pre-trained LM head. Let $N_a$ denote the number of action-related tokens per step and $D_v$ the vocabulary size. Since we perform this projection at every layer except the last (where we don't need to reinject at the next layer as it is just the last year), the additional FLOPs are:

$$\text{FLOPs}_{\text{PROJ}} = (L - 1) \cdot 2N_a D D_v. \tag{18}$$

**Reinjection Cost.** When triggered, UAOR acts as an additional FFN-like module comprising a retrieval operation. It involves two linear transformations (Query-Key and Attention-Value) with

shared weights. Let $N_o$ be the number of observation tokens. The cost for a single reinjection is $\text{FLOPs}_{\text{SINGLE\_INJ}} \approx 4NN_oD$. Assuming the reinjection is triggered in $L_\gamma$ layers (where uncertainty $u > \gamma$), the total reinjection cost is:

$$\text{FLOPs}_{\text{INJ}} = L_\gamma \cdot 4NN_oD. \tag{19}$$

**Total Overhead Ratio.** We quantify the additional computational burden using the ratio $r_{\text{cost}}$:

$$r_{\text{cost}} = \frac{\text{FLOPs}_{\text{PROJ}} + \text{FLOPs}_{\text{INJ}}}{\text{FLOPs}_{\text{LM}}} \approx \underbrace{\frac{(L-1) \cdot 2N_aDD_v}{L \cdot (24ND^2 + 4N^2D)}}_{\text{Projection term}} + \underbrace{\frac{L_\gamma \cdot 4NN_oD}{L \cdot (24ND^2 + 4N^2D)}}_{\text{Reinjection term}}. \tag{20}$$

Note that we approximate the denominator by dominating term $24ND^2$ (since $D \gg N$) for clarity. Simplifying the terms yields:

$$r_{\text{cost}} \approx \frac{N_aD_v}{12ND} + \frac{L_\gamma}{L} \cdot \frac{N_o}{6D}. \tag{21}$$

**Case Study.** We analyze the overhead for two representative VLA models, OpenVLA-OFT (Kim et al., 2025a) and CogACT (Li et al., 2024a), using the Llama-2-7B backbone ($D = 4096, D_v = 32000$).

- **OpenVLA-OFT**: With sequence length $N \approx 600$ and action tokens $N_a = 56$, the projection overhead is $\approx \frac{56 \times 32000}{12 \times 600 \times 4096} \approx \textbf{6.0\%}$. On LIBERO-Long, the statistical trigger rate is $\frac{L_\gamma}{L} \approx 20\%$. With observation tokens $N_o = 513$, the reinjection overhead is $0.2 \times \frac{513}{6 \times 4096} \approx \textbf{0.4\%}$. The total overhead is roughly **6.4%**.
- **CogACT**: With $N \approx 300$ and $N_a = 1$ (predicting one condition token per step), the projection overhead drops significantly to $\approx \frac{1 \times 32000}{12 \times 300 \times 4096} \approx \textbf{0.2\%}$. Assuming a similar trigger rate, the total overhead remains negligible at $< 1\%$.

This analysis confirms that UAOR is computationally efficient, particularly for those VLA models who generate one action-related token per step, and introduces minimal latency compared to the heavy backbone computation.

# E  THE USE OF LARGE LANGUAGE MODELS (LLMS)

In this paper, we use large language models (LLMs), such as ChatGPT, to assist with writing refinement, grammar correction, formatting, and preliminary literature search during manuscript preparation.

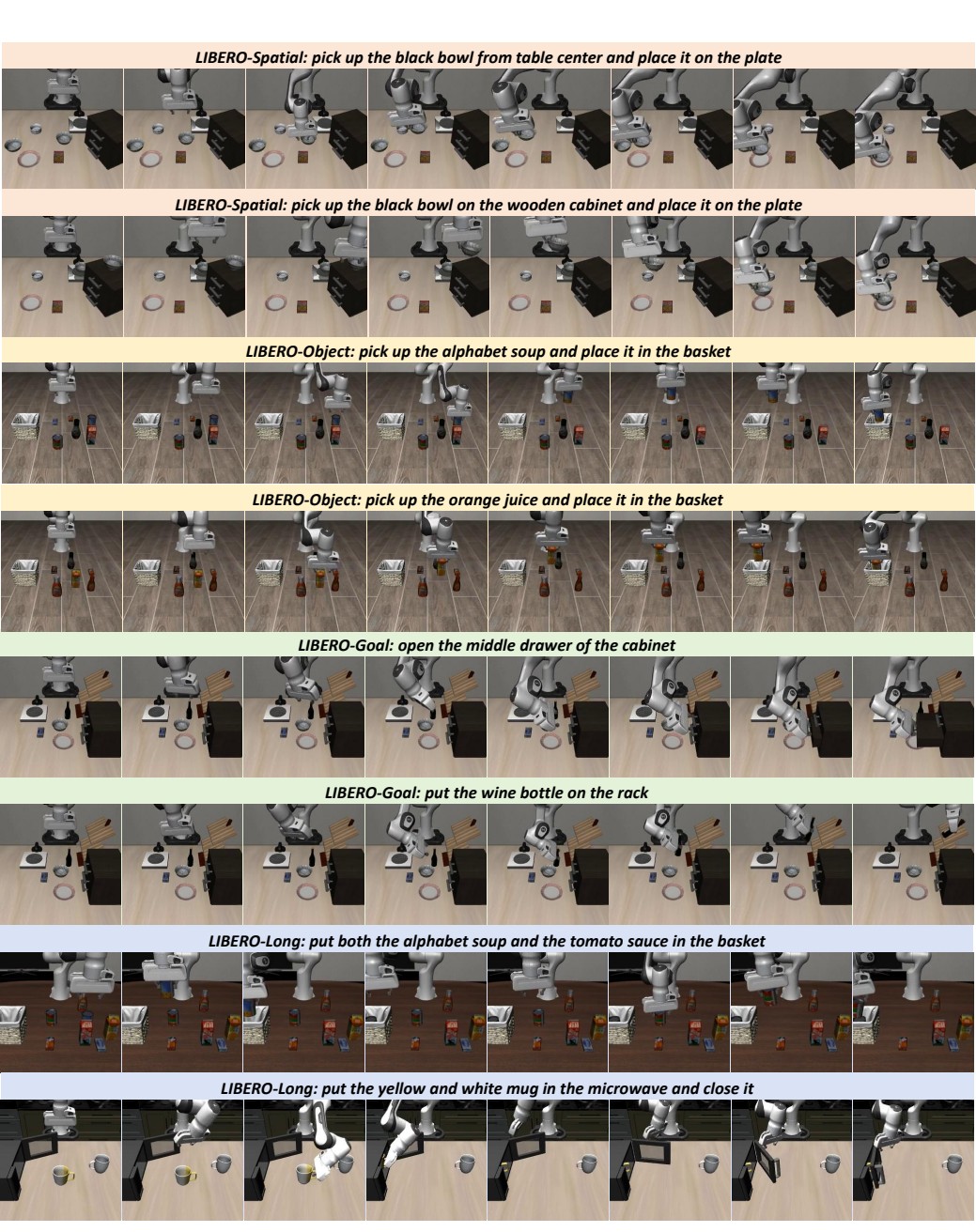

Figure 9: **Manipulation Visualizations in the LIBERO Simulation Environment.** We present the execution processes of OpenVLA-OFT with UAOR across LIBERO-Spatial, LIBERO-Object, LIBERO-Goal, and LIBERO-Long, demonstrating its strong performance under diverse instructions and a wide range of tasks. Each row shows a temporally ordered sequence from left to right.

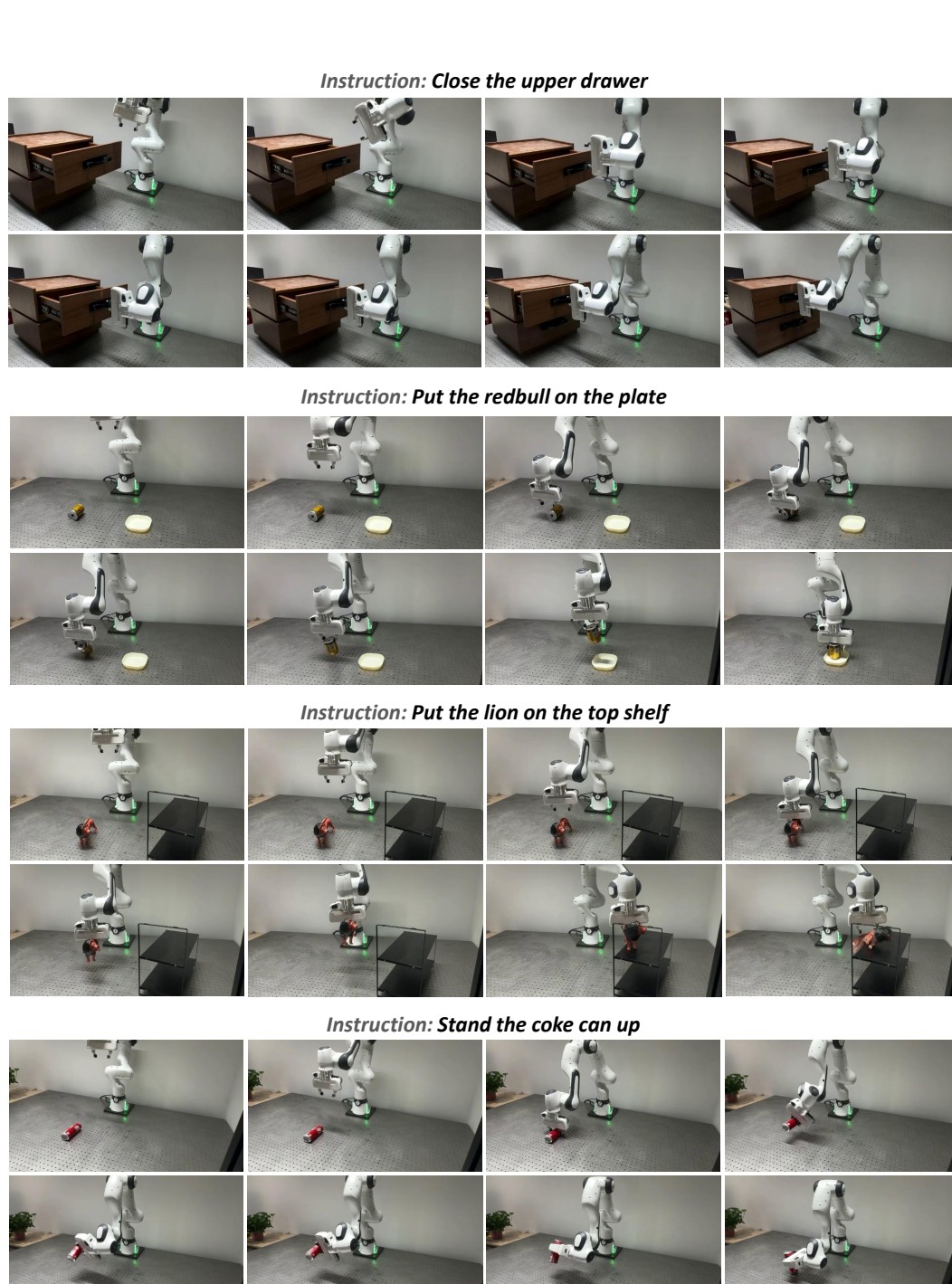

Figure 10: **Manipulation Visualizations in the Real-World Environment.** We present the execution processes of OpenVLA-OFT with UᴀOR across four real-world tasks, demonstrating its strong effectiveness and practicality in real-world scenarios. Each pair of rows shows a temporally ordered sequence from left to right.

