# OpenReview forum: "UAOR: Uncertainty-aware Observation Reinjection for Vision-Language-Action Models"
_ICLR.cc/2026/Conference — ICLR 2026 Conference Withdrawn Submission_

### Official Review · Reviewer_rnbV · 2025-10-27

**Soundness:** 3
**Presentation:** 4
**Contribution:** 3
**Rating:** 4
**Confidence:** 4

**Summary:**

This paper introduces Uncertainty-aware Observation Reinjection (UAOR), a novel, training-free, and plug-and-play module designed to enhance the performance of Vision-Language-Action (VLA) models. The core idea is to counteract the "forgetting" of initial observation information as data propagates through the model's layers. The authors propose a metric, "Action Entropy," to measure layer-wise uncertainty during inference. When this uncertainty surpasses a predefined threshold at a given layer, UAOR "reinjects" the original observation features into the Feed-Forward Network (FFN) of the subsequent layer. This mechanism is inspired by the concept of FFNs acting as key-value memory, allowing the model to dynamically "re-focus" on crucial sensory inputs when its confidence wanes. The authors provide a theoretical analysis based on information theory to justify their approach and validate its effectiveness through extensive experiments on multiple simulation benchmarks (LIBERO, SIMPLER, CALVIN) and in the real world, demonstrating consistent performance improvements across various VLA architectures with negligible computational overhead.

**Strengths:**

- The work is motivated by a clear and compelling intuition—that observation information decays through deeper network layers, leading to increased uncertainty.
- The authors demonstrate the effectiveness of UAOR across three different VLA baselines with varying architectures (single-system and dual-system) and scales (0.5B to 7B), on three distinct simulation benchmarks.
- The inclusion of a theoretical analysis (Section 3.4) adds significant depth and credibility to the paper.
- The paper is exceptionally well-written, with a logical flow and clear explanations of complex concepts.

**Weaknesses:**

- The proposed "Action Entropy" metric relies on projecting the hidden states of every intermediate layer through the MLP of the final layer to get a probability distribution. This approach feels somewhat ad-hoc and potentially inefficient. It raises questions about whether this is the most direct or optimal way to measure uncertainty, as it depends on a component (the final layer's MLP) far downstream from where the uncertainty is being measured.

- The rationale for this one-layer delay is not discussed. Ablation studies exploring reinjection into the current layer's FFN, or even into the self-attention block, are missing and would provide valuable insight into this specific design choice.

- While the selection of baselines is commendably diverse, the paper acknowledges that it does not apply UAOR to some of the most recent and powerful state-of-the-art models like $\pi_0$. Demonstrating gains on these near-SoTA models would make the claims of general applicability even more powerful.

- The analysis in Section 4.4 explores a heuristic for weighting visual tokens based on language similarity but finds it does not improve performance over uniform weighting. While the authors provide plausible hypotheses, this result is somewhat counter-intuitive and warrants a deeper investigation. It might suggest that the simple language-similarity weighting is flawed, rather than the concept of weighting itself.

**Questions:**

- Could you provide a more detailed rationale for the "Action Entropy" metric design? Specifically, why project intermediate hidden states through the final layer's action head, rather than using a more direct uncertainty measure from the hidden states themselves (e.g., entropy of the feature distribution, or a lightweight learned uncertainty head)? What is the computational overhead of these repeated projections during inference?

---

> ### Author Response · Authors · 2025-11-26
> **Response to Reviewer rnbV Part 1**
>
> Dear Reviewer rnbV，
>
> We sincerely thank you for your encouraging feedback and for recognizing the **clear and compelling intuition** behind our work regarding observation decay. We are particularly gratified by your appreciation of our **comprehensive empirical validation** across varying VLA architectures (single/dual-system) and scales, as well as the **significant depth and credibility** provided by our theoretical analysis. We also value your positive remarks on the **exceptional clarity and logical flow** of our presentation. Below, we respectfully address your questions and concerns in detail.
>
> ---
>
> > ### **W1 & Q1: Rationale, Validity, and Overhead of Action Entropy**
>
> Thanks for your insightful comment. We have provided a comprehensive answer to these questions in **Concern 2** of the **General Response 3-5**. We respectfully invite you to refer to that section.
>
> ---
>
> > ### **W2: Rationale for "One-Layer Delay" and injection location**
>
>
> We thank the reviewer for probing into these design choices. To provide empirical evidence, we conduct a new ablation study comparing injection at different **Timings** (Current vs. Next Layer) and **Modules** (Self-Attention vs. FFN) **(added into the new Appendix C.1)**.
>
> **Table 1: Ablation on Injection Location and Timing (OpenVLA-OFT on LIBERO)**
>
> |Injection Timing|Injection Module|Spatial|Object|Goal|Long|Avg.|Latency|Overhead|
> |:-|:-|:-:|:-:|:-:|:-:|:-:|:-:|:-:|
> |-|*Baseline (No Injection)*|98.2|98.2|97.6|94.2|97.1|0.161s|-|
> |**Current Layer ($\ell$)**|Self-Attention (SA)|98.2|98.0|97.8|95.8|97.5|0.195s|+21.1%|
> |**Current Layer ($\ell$)**|Feed-Forward (FFN)|98.6|98.2|98.0|95.8|97.7|0.182s|+13.0%|
> |**Next Layer ($\ell+1$)**|Self-Attention (SA)|98.4|98.0|97.8|94.8|97.3|0.170s|+5.6%|
> |**Next Layer ($\ell+1$)**|**Feed-Forward (UAOR)**|**99.0**|**98.4**|**98.2**|**96.2**|**98.0**|**0.169s**|**+5.0%**|
>
> **Analysis of Results:**
>
> **1. Why "One-Layer Delay"? (Effectiveness & Efficiency)**
>
> *   **Effectiveness:** As shown in Table 1 injecting into the *Current FFN* (97.7%) and *Next FFN* (98.0%) yields comparable performance. This is because the underlying operation is mathematically identical (using the FFN's input to retrieve observation features and blending them with the output). The slight edge for *Next Layer* may stem from using a more processed hidden state as the query.
> *   **Efficiency:**
>     *   *Current Layer SA (+21.1%):* Modifying the SA output changes the input to the subsequent FFN. This forces the model to **re-compute the entire FFN** for that layer, causing the highest latency (0.195s).
>     *   *Current Layer FFN (+13.0%):* While we don't need to re-compute the FFN matrix multiplication, we must **fetch the cached FFN input** from memory (if we don't want to backtrack) to perform retrieval and then blend it with the output. This extra Memory I/O and pipeline interruption incurs noticeable overhead (0.182s) compared to the streamlined Next Layer injection (+5.0%).
>     *   *Next Layer FFN (Ours, +5.0%):* By injecting into the next layer, we seamlessly integrate the retrieval into the forward pass without backtracking or memory overhead, achieving the optimal efficiency.
>
> **2. Why FFN over Self-Attention?**
> Comparing *Next Layer FFN* (98.0%) with *Next Layer SA* (97.3%) confirms FFN is the superior injection site.
>
> *   **Reason:** FFNs structurally function as **Key-Value Memories** [1] [2], making them the natural component for retrieving and storing external information (observation). In contrast, Self-Attention focuses on token-to-token contextualization; injecting external features there may dilute the attention distribution, leading to slightly inferior performance.
>
> [1] Geva et al. "Transformer Feed-Forward Layers Are Key-Value Memories". *EMNLP*, 2021.
>
> [2] Jie  et al. "Memory-space visual prompting for efficient vision-language fine-tuning." *ICML*, 2024

---

> ### Author Response · Authors · 2025-11-29
> **Response to Reviewer rnbV Part 2**
>
> > ### **W3: Application to recent SOTA models (e.g., $\pi_0$)**
>
> We appreciate this suggestion to strengthen our claims of general applicability. We have addressed this in **Concern 3** of the **General Response 6**.
>
> *   **New Results:** We applied UAOR to the state-of-the-art flow-matching policy **$\pi_0$** (Black et al., 2024).
> *   **Findings:** As shown in **Table 5 (General Response 6)**, UAOR consistently boosts $\pi_0$ performance across all LIBERO suites (Average **+1.5%**), with a notable **+2.0%** gain on Long-horizon tasks. This confirms that UAOR remains highly effective even on top of cutting-edge architectures.
>
> ---
>
> > ### **W4: Counter-intuitive results of visual weighting; suggestion for deeper investigation**
>
> We agree that this result warrants deeper investigation. To address this, we extended our analysis to include **Proprioception-guided Weighting** (visualizing similarities between visual tokens and proprioceptive state tokens).
>
> * **New Results:** As detailed in the updated Section 4.4 and **Table 7** in the revised manuscript, Proprio-guided weighting (96.9%) also fails to outperform uniform weighting (98.0%) and performs even worse than Language-guided weighting (97.3%).
>
> *   **Deeper Insight:** We hypothesize that the issue lies in the **quality of the heuristic alignment**:
>
>     * **Misalignment:** Pretrained VLMs (like CLIP/SigLIP) align text and images globally, but fine-grained token-level alignment is often noisy. Proprioception (joint angles) lacks any spatial correspondence with visual pixels, making heuristic weighting underperform (as seen in the diffuse patterns in **Figure 6b**).
>
>     * **Disruption:** VLA models are trained under uniform token weighting. Imposing an inaccurate external bias (heuristic weighting) may disrupt the model's learned internal attention mechanism.

---

### Official Review · Reviewer_A2fr · 2025-10-28

**Soundness:** 3
**Presentation:** 3
**Contribution:** 2
**Rating:** 6
**Confidence:** 3

**Summary:**

This paper proposes UAOR, a lightweight, training-free module designed to boost VLA models. It aims to enable VLA models
look more clearly on the observation during inference, enabling more confident and faithful action generation.

**Strengths:**

1. Demonstrates a strong understanding of the current limitations of VLA models.

2. Attempts to address the identified problems in a general and systematic way.

3. Proposes the concept of action entropy and applies it effectively in the forward process. The accompanying theoretical analysis is solid and convincing.

4. Provides a well-designed ablation study to examine the effects of observation injection.

**Weaknesses:**

1. The explanation of why forgetting leads to uncertainty lacks a clear reasoning process.

2. Theorems 3.1–3.4 appear largely independent and not well integrated. It would be better to unify them into a holistic framework or justify that they represent complementary perspectives on the same problem.

3. Real-world experiments are only conducted using Open-VLA, neglecting other baseline models such as CogACT.

4. The relationship between α and γ should be jointly analyzed, as variations in one may influence the behavior or trend of the other.

**Questions:**

1. Is the action entropy gate essential for injecting observation features?

2. Can hidden states or proprioceptive states reliably represent action entropy?

3. In Equation (9) of Algorithm 1, why is h_t^{(l+1)} used instead of h_t^{(l)}? The paper does not discuss uncertainty propagation between adjacent layers.

4. In Section 4.4, the discussion on token-level weighting explores applying weights to visual tokens based on language instructions. However, the ablation study in Section 4.3 shows that injecting language instructions offers no performance gain. This suggests that the similarity between visual and language tokens provides limited benefit. Why not explore similarities between visual and proprioceptive tokens instead? Besides, It is plausible that redundant (similar) information adds little value, whereas orthogonal (complementary) information may be more beneficial.

---

> ### Author Response · Authors · 2025-11-26
> **Response to Reviewer A2fr Part 1**
>
> Dear Reviewer A2fr，
>
> We sincerely thank you for your constructive feedback and for recognizing our **strong understanding of current VLA limitations**. We are particularly encouraged by your positive assessment of our work as a **general and systematic attempt** to address these issues, as well as your appreciation for the **solid theoretical analysis** underlying our proposed action entropy concept. We also value your recognition of our **well-designed ablation study** regarding observation injection. Below, we respectfully address your questions and concerns in detail.
>
> ---
>
> > ### **W1: The explanation of why forgetting leads to uncertainty lacks a clear reasoning process**
>
> Thanks for your insightful comment.  Since this is also a major concern shared by other reviewers, we have provided a comprehensive and detailed explanation in the **Concern 1** section of **General Response 2**. We kindly refer you to it for more details.
>
> ---
>
> > ### **W2: Theorems 3.1–3.4 appear largely independent and not well integrated**
>
> We thank the reviewer for this constructive suggestion. We apologize if the logical connections between the theorems were not explicitly articulated in the initial submission.
>
> We respectfully clarify that these four theorems form a **sequential and causal logical chain** that theoretically grounds every aspect of UAOR. To make this explicit, we have **rewritten the concluding paragraph of Section 3.4** (now titled **"Theoretical Integration"** in the revised paper) to present them under a **unified logical framework**:
>
> 1.  **Theorem 3.1 (The Mechanism):** Establishes the foundational premise—that our reinjection mechanism mathematically guarantees the restoration of observation information ($I(\hat{h}; o) \uparrow$).
> 2.  **Theorem 3.2 (The Effect):** Links the mechanism to the outcome. It proves that the information gain from Thm 3.1 directly leads to reduced action uncertainty ($H(y|\hat{h}) \downarrow$).
> 3.  **Theorem 3.3 (The Objective):** Justifies the *validity* of this operation via the Information Bottleneck principle. It ensures that the reinjected features are not merely adding noise or redundancy (which increases representation cost), but are contributing significant predictive gains that outweigh the cost, thereby optimizing the trade-off between feature compression and action prediction.
> 4.  **Theorem 3.4 (The Control):** Justifies our trigger strategy. It proves that triggering reinjection based on high uncertainty (Entropy > $\gamma$) maximizes the expected relevance of the injected information compared to indiscriminate injection.
>
> We hope this revision can clearly demonstrate how the theorems collectively constitute a holistic theoretical basis for our method.
>
> ---
>
> > ### **W3: Real-world experiments are only conducted using OpenVLA-OFT**
>
> We sincerely thank the reviewer for this valuable suggestion. We agree that validating our method on a broader range of baselines in the real world strengthens our claims of generalizability.
>
> In response, we have conducted **additional real-world experiments using the CogACT baseline** across the same four manipulation tasks. The results, now updated in **Figure 4** of the revised manuscript, demonstrate that UAOR consistently enhances the performance of CogACT as well. specifically:
>
> * **Universal Improvement:** UAOR improved the success rate of CogACT in **all four tasks**.
>
> * **Significant Gains:** The average success rate of CogACT increased from **57.5%** to **70.0%** with the integration of UAOR . Notably, in the *Put the redbull on the plate* task, UAOR improved the success rate by **20%** (absolute).
>
> These new results confirm that our method is not specific to OpenVLA-OFT but acts as a robust, plug-and-play module capable of enhancing various VLA architectures in complex real-world environments.

---

> ### Author Response · Authors · 2025-11-26
> **Response to Reviewer A2fr Part 2**
>
> > ### **W4: The relationship between α and γ should be jointly analyzed**
>
> We sincerely thank the reviewer for this insightful suggestion. We agree that the blending factor $\alpha$ and the uncertainty threshold $\gamma$ are coupled, and analyzing them in isolation might overlook their interaction effects.
>
> To address this, we have conducted a **joint sensitivity analysis** on LIBERO-Long based on OpenVLA-OFT and added a **3D surface plot (Figure 5c)** in the revised manuscript to visualize the combined impact of $\alpha$ and $\gamma$ on the average success rate.
>
> * **Interaction Confirmation:** As shown in Figure 5c, the performance landscape forms a convex surface with a distinct peak, confirming that optimal performance requires balancing both parameters simultaneously.
>
> * **Trade-off Analysis:** We observe that extreme combinations degrade performance. For instance, a low $\gamma$ (frequent injection) combined with a high $\alpha$ (strong blending) leads to a significant drop, likely because the model's internal representations are excessively overwritten. Conversely, a high $\gamma$ with a low $\alpha$ results in insufficient correction.
>
> * **Robustness:** The "sweet spot" (yellow region in Figure 5c) is relatively broad, indicating that UAOR is robust within a reasonable range of parameter settings, rather than being extremely sensitive to precise tuning.
>
> ---
>
> > ### **Q1 & Q2: Is the action entropy gate essential? Can hidden states or reliably represent action entropy?**
>
> We thank the reviewer for these critical questions regarding the necessity and validity of our trigger mechanism. As these questions address the core rationale of our metric design—a topic shared by multiple reviewers—we have provided a comprehensive and detailed answer in the **Concern 2** section of the **General Response 3-5**. We respectfully invite you to refer to that section. (Note: we use the hidden states of the action-related tokens to compute action entropy which don't include the proprioceptive states, the proprioceptive states are one part of the observation tokens if available.)
>
> ---
>
> > ### **Q3: Is the action entropy gate essential? Can hidden states or reliably represent action entropy?**
>
> We thank the reviewer for pointing out this notational detail. We clarify that the indexing in Algorithm 1 reflects the **sequential nature of the inference process (We kindly refer you to combine Figure 3 for better understanding)**:
>
> 1.  **Detection at Layer $\ell$:** We first compute the action entropy and uncertainty $u_t^{(\ell)}$ using the FFN's output hidden states $\tilde{\boldsymbol{h}}_t^{(\ell)}$ at layer $\ell$.
> 2.  **Intervention at Layer $\ell+1$:** **Since the forward pass at layer $\ell$ has already completed, we cannot modify it retrospectively.** Therefore, if high uncertainty is detected ($u_t^{(\ell)} > \gamma$), we trigger the observation reinjection at the **immediate next layer**, i.e., layer $\ell+1$.
> 3.  **Query Selection:** Consequently, we use the hidden states input to the FFN at layer $\ell+1$ (denoted as $\boldsymbol{h}_t^{(\ell+1)}$) as the query to retrieve observation features. Then we reinject the retrieve observation features to the original FFN's output hidden states of  layer $\ell+1$.
>
> We apologize that the generic notation $\ell$ used in the original Equations 8 and 9 (describing the operation in isolation) caused confusion when cross-referenced with the Algorithm. **We have revised the main text and equations (Section 3.3)** to explicitly use indices $\ell$ (for uncertainty) and $\ell+1$ (for injection) to ensure consistency with Algorithm 1 and Figure 3.
>
> ---
>
> > ### **Q4: Explore similarities between visual and proprioceptive tokens**
>
> We thank the reviewer for this insightful suggestion. In response, we conducted additional experiments using **proprioceptive states** to weight visual tokens (visualizing in the newly added **Figure 6b**).
>
> * **Results:** As shown in the updated **Table 7** in the revised paper, weighting based on proprioception (96.9%) performs slightly worse than uniform weighting (98.0%) and language-based weighting (97.3%).
> * **Analysis:** Visualizations (Figure 6b) reveal that unlike language, proprioceptive tokens result in **diffuse and semantically ambiguous attention maps**. This is likely because proprioceptive features (joint states) lack explicit spatial alignment with visual features in the pre-trained embedding space. Consequently, using them for weighting introduces noise, disrupting the model's reasoning. This reinforces our conclusion that **uniform reinjection** is the most robust training-free strategy, as it allows the model's internal attention mechanism to function without biased interference.

---

### Official Review · Reviewer_7dgg · 2025-10-31

**Soundness:** 3
**Presentation:** 3
**Contribution:** 2
**Rating:** 4
**Confidence:** 5

**Summary:**

This paper proposes UAOR (Uncertainty-Aware Observation Reinjection), a training-free and plug-and-play module designed to enhance VLA models. UAOR measures layer-wise action entropy to detect high-uncertainty regions during inference. When uncertainty exceeds a threshold, UAOR reinjects observation features into the next layer’s FFN, treating the module as a key-value memory to restore visual representation. Experiments across multiple benchmarks and real-world robot tasks show a certain degree of performance gains with negligible computational overhead.

**Strengths:**

S1. I think the problem addressed by this paper is very important, as overly deep LLM layers do tend to ignore certain visual information.

S2. The paper proposes a simple yet efficient method to alleviate this issue.

**Weaknesses:**

W1. My main concern lies in whether using Action Token Entropy to measure visual uncertainty is reasonable.


W2. Pretrained VLA models typically generate actions only from the final layer, without utilizing or supervising intermediate features for action prediction. Therefore, the observed layer-wise “action” token entropy may result from the training paradigm itself rather than reflecting the actual dynamics of feature changes within the model. I suggest that the authors finetune the VLA model so that each LLM layer outputs actions for verification.

W3. Regarding the analysis of Action Token Entropy in Figure 1, I also find it unconvincing. Prior machine learning studies have shown that deeper layers’ feature representations are more task-specific, which could naturally cause the entropy variation (rather than the reason proposed by the authors).

**Questions:**

Q1. My minor concern is that the performance improvements brought by UAOR are relatively small. For example, on LIBERO and CALVIN, the gains are only about 0.9\%, which seems quite incremental. I therefore recommend that the authors consider finetuning the model so that it can better adapt to the newly introduced token sequence.


Q2. Since the authors’ motivation is very good, I would be willing to raise my score if they can address my concerns.

---

> ### Author Response · Authors · 2025-11-26
> **Response to Reviewer 7dgg Part 1**
>
> Dear Reviewer 7dgg，
>
> We sincerely thank you for your constructive feedback and for recognizing the **importance of the problem** regarding visual information loss in deep LLM layers. We are particularly encouraged by your appreciation of our proposed method as a **simple yet efficient** solution to alleviate this issue. Below, we respectfully address your questions and concerns in detail.
>
> ---
>
> > ### **W1: Whether using Action Token Entropy to measure visual uncertainty is reasonable**
>
> Since this is also a major concern shared by multiple reviewers, we have presented a comprehensive and detailed clarification in the **Concern 2** section of the **General Response 3-5**. We sincerely hope this can address your concern.
>
> ---
>
> > ### **W2 & W3: The analysis of action entropy and suggestion for fine-tuning the VLA models for verification**
>
> Thanks for your insightful comment and the constructive suggestion for verification. We acknowledge that the phenomenon of action entropy may be inherent to the training paradigm. However, please kindly refer to **General Response 2 (Concern 1)**, where we provide comprehensive and detailed analysis including empirical evidence (Layer-wise Attention Analysis) for it.
>
> We have fully implemented your suggestion to **fine-tune the VLA model** (specifically, training linear lm heads at intermediate layers). We sincerely refer you to the **Concern 2** section in **General Response 3-5**, where we present the quantitative results of this experiment. The findings confirm that intermediate layers do encode significant action semantics, validating the rationale of our metric.
>
> ---
>
> > ### **Q1: The performance improvements brought by UAOR are relatively small and suggestion for fine-tuning the model**
>
> We thank the reviewer for this constructive suggestion. As for the concern on the performance gains, we have presented a detailed clarification in the **Concern 3** section of **General Response 6**, which we hope can address your concern. Following your suggestion for **fine-tuning**, we have provided a detailed discussion regarding the trade-offs between training-free efficiency and fine-tuning adaptation, along with verification experiments, in the **Concern 2** section of **General Response 3-5**. We respectfully invite you to refer to these sections for more details.

---

### Official Review · Reviewer_surj · 2025-10-31

**Soundness:** 2
**Presentation:** 3
**Contribution:** 2
**Rating:** 4
**Confidence:** 4

**Summary:**

This paper introduces a training-free method for reinjecting observations back to transformer layers of VLAs. The paper claims that VLAs tend to progressively forget observations, and they proposes to blend the observation features with the FFN output via the key-value memory mechanism. The paper compares against other VLAs on the LIBERO, SIMPLER, and real-world benchmarks, showing some gains while not introducing too much latency.

**Strengths:**

- The paper is fairly well-written and easy to understand.
- The problem studied is important.

**Weaknesses:**

- The core premise of the paper is not well-supported. The author claims that: "Our key intuition is that after ingesting the observation, the model tends to progressively “forget” during forward inference" and back this up by Figure 1, where the **early** layers of the VLA experiences a mild increase in action uncertainty. This is neither convincing nor well-explained. To me, Fig. 1 is actually quite reasonable: early in the computation, there's more uncertainty in the action distribution but as the model has the chance to process the representations more in later layers, the uncertainty decreases, almost to 0 at the last layer.
	- Even if the claim that increasing in uncertainty equals to forgetting observation, this doesn't explain how the uncertainty eventually decreases at the mid layer and almost to 0 at the final layer.
	- It is not clear that action uncertainty can be tied to "forgetting observation" as neural networks dynamics can be complicated. We need stronger evidence, for example drawing tools from interpretability research, perhaps looking at some sort of cross attention between the outputs and the observations, to validate the core premise.
- The quantitative results are not clear.
	- All the gains are very modest.
	- No measure of statistical confidence. No details on how many trials were run.
	- Table 4 is not clear. What task / benchmark is this?

- Some technical details are not well-justified.
	- It is not clear that if you just take the middle hidden representations and decode that, computing the "action" entropy is still meaningful. For intuition, for classifer guidance in image diffusion literature, you can't just take the intermediate results during denoising, and ask a classifer that was ever only trained on clean images to infer on noisy inputs. You either need to train the classifier on noisy images too or use techniques like diffusion posterior sampling, tangent projection et cetera. So again, it's not clear to me that decoding actions from an intermediate layer is a meaningful operation.
	- If I understand this correctly, the paper applies their re-injection during test-time without ever training the base policy. During training, the base model is only exposed to hidden representations while during inference it must deal with "out-of-distribution" embeddings from reinjecting observation. It's not clear why this OOD issue wouldn't destroy performance.

**Questions:**

- Do you see any connection between the problem and residual networks? The intuition of skip connection is fairly similar: as the depth increases, it's harder to train, and the network "forgets" its observation. Thus, skip connection adds the observation back after each layer.
- It would be nice to see a conceptual and also quantitative discussion / comparison with skip connections (which is actually already present in the transformer architecture)

---

> ### Author Response · Authors · 2025-11-26
> **Response to Reviewer surj Part 1**
>
> Dear Reviewer surj，
>
> We sincerely appreciate your detailed and valuable feedback. In response to the concerns raised, we provide clarifications and additional analysis below. Hope they can address your concerns.
>
> ---
>
> > ### **W1: The core premise of the paper is not well-supported**
>
> We thank the reviewer for this insightful comment. As this was a major concern shared by multiple reviewers, we have provided a comprehensive and detailed answer in the **Concern 1** section of **General Response 2**.
>
> ------
>
> > ### **W2: The quantitative results are not clear**
>
> We thank the reviewer for scrutinizing our experimental results. We would like to address the three specific points raised:
>
> **1. On the magnitude of performance gains**
>
> Since this is also a general concern shared by multiple reviewers, we have presented a comprehensive and detailed clarification in the **Concern 3** section of **General Response 6**. Hope this can address your concern.
>
> **2. On statistical confidence and trial details**
>
> We apologize for the oversight in not explicitly detailing the experimental settings in the main text of the initial submission. **To clarify, the main experiments were conducted using three different random seeds to ensure reliability.** In fact, a detailed multi-seed evaluation on the LIBERO benchmark based on OpenVLA-OFT was already included in **Appendix C.1 of the original submission** (now **omitted to avoid redundancy**), where UAOR demonstrated high stability. To provide a comprehensive and clear measure of statistical confidence, we have **updated the main tables** in the revised manuscript to explicitly report the mean and standard deviation for **all baseline models as well as their UAOR-enhanced counterparts**.
>
> **3. On the clarity of Table 4**
>
> We apologize for the confusion caused by the missing caption details. **Table 4** (**Now Table 6 in the revised paper** ) presents the **Ablation Study on Reinjection Information** (comparing different injection contents like visual, proprioception and instruction features) on the **LIBERO benchmark** based on the **OpenVLA-OFT** backbone. We have revised the caption in the updated manuscript to clearly reflect this.
>
> ---
>
> > ### **W3: Some technical details are not well-justified**
>
> Thank you for raising these technical concerns. We clarify our design choices and how we mitigate the issues you pointed out.
>
> **1. On the validity of decoding intermediate layers (Action Entropy)**
>
> Since this is also a major concern shared by multiple reviewers, we have presented a comprehensive and detailed clarification in the **Concern 2** section of the **General Response 3-5**.
>
> **2. On the OOD concern regarding inference-time injection:**
> We understand the concern that modifying hidden states might push representations out of distribution. However, our method is designed to **steer** the model back to the correct manifold rather than disrupting it.
>
> - **Re-grounding vs. OOD:** The "forgetting" phenomenon itself pushes the model towards an ungrounded (hallucinated) state. UAOR acts as a correction mechanism. By retrieving and blending original observation features (which are generated by the *same* frozen encoders and are thus compatible with the latent space), we are effectively "reminding" the model of the observation context.
> - **Soft Blending:** We do not brutally replace the hidden states. Instead, we use a blending mechanism (controlled by a small hyperparameter $\alpha$ (e.g., 0.05)) via the FFN's key-value retrieval structure. This ensures the modification is a subtle "nudge" rather than a shock.
> - **Analogy to Representation Engineering:** Our approach aligns with the emerging field of **"Activation Steering"** or **"Representation Engineering"** [1] [2] [3], where inference-time interventions on hidden states have been proven effective in controlling model behavior (e.g., attribute control).
> - **Empirical Robustness:** In our extensive hyperparameter sweeps (see Figure 5 in our revised paper), we observed that the performance **typically improves or remains robust**, without exhibiting the instability or collapse often associated with OOD perturbations. This empirically validates the safety of our operation.
>
> [1] Zou et al. "Representation Engineering: A Top-Down Approach to AI Transparency." *arXiv*, 2023.
>
> [2] Xu et al. "Word Embeddings Are Steers for Language Models." *ACL*, 2024.
>
> [3] Konen et al. "Style Vectors for Steering Generative Large Language Models". *EACL*, 2024

---

> ### Author Response · Authors · 2025-11-26
> **Response to Reviewer surj Part 2**
>
> > ### **Q1 & Q2: Connection to residual networks / skip connections, and comparison**
>
> We thank the reviewer for drawing this insightful connection. We agree that the intuition behind UAOR shares similarities with Residual Networks (ResNets): both aim to mitigate signal decay (or "forgetting") as network depth increases. However, standard residual connections primarily address **gradient flow and optimization**, whereas UAOR addresses **cross-modal grounding and specific information retrieval**. Below, we provide a detailed comparison on conceptual, mechanistic, and quantitative levels.
>
> **1. Conceptual Comparison: Optimization vs. Grounding**
>
> - **Standard Skip Connections (Local & Indiscriminate):**
>   - **Mechanism:** As established in Transformer architectures, standard skip connections are **local and intra-block** (e.g., adding the input of a Multi-Head Attention or FFN block to its output). Their primary goal is to alleviate **vanishing gradients** during training.
>   - **Limitation:** Crucially, these connections propagate the *entire* hidden state indiscriminately. As shown in our analysis of OpenVLA-OFT (**Figure 2**), despite the presence of these local residuals, the attention to **observation tokens** still diminishes significantly in deeper layers. This suggests that standard residuals simply propagate the current state—even if it has already suffered from **observation forgetting** or information dilution—without specifically refreshing the observation signal.
> - **UAOR (Global & Targeted):**
>   - **Mechanism:** In contrast, UAOR acts as a **Global Observation-to-Action connection**. It does not just pass the adjacent layer's state forward; it explicitly retrieves features from the **original observation memory** (the initial observation tokens).
>   - **Goal:** It targets **inference-time information dilution**. By injecting these "fresh" observation features directly into the deep layers, UAOR acts as a specialized channel to restore cross-modal dependency, counteracting the potential forgetting observed in the baseline.
>
> **2. Mechanistic Comparison: Static Identity vs. Dynamic Retrieval**
>
> * **Static vs. Adaptive:** Standard skip connections are **static and unconditional**—they operate at every sub-layer. UAOR is **dynamic and conditional**. We only trigger injection at layers with high action entropy ($u^{\ell}_{t} > \gamma$), effectively creating an **"on-demand" skip connection**.
> * **Identity vs. Retrieval:** Skip connections perform an identity mapping (simple addition). UAOR employs an **FFN-like key-value retrieval**: it uses the current hidden state as a query to selectively extract relevant observation clues. This ensures that we inject *useful* information compatible with the current context, rather than blindly adding raw features.

---

> ### Author Response · Authors · 2025-11-26
> **Response to Reviewer surj Part 3**
>
> > ### **Q1 & Q2: Connection to residual networks / skip connections, and comparison**
>
> **3. Quantitative Discussion**
>
> We agree that UAOR conceptually functions as a refined form of skip connection. To rigorously address the reviewer’s request for a **quantitative comparison**, we conducted a comprehensive **$2 \times 3$ factorial ablation study** (which has been added to **Section 4.3: Ablation Study** in the revised manuscript). We compared our method against **"Mean-Residual"** and **"Mean-Blending"** baselines. Since the observation tokens and hidden states differ in sequence length, element-wise addition (standard ResNet) is impossible. Therefore, we aggregate observation features via **Mean Pooling** for the residual baselines.
>
> **(1) Experimental Formulation**
>
> *   **Mean-Residual:** Directly adds the mean-pooled observation features to the hidden state ($h' = h + o_{mean}$) . Represents a naive residual connection.
> *   **Mean-Blending:** Blends the mean-pooled observation features using $\alpha$ ($h' = (1-\alpha)h + \alpha o_{mean}$). Represents a "softer" residual.
> *   **UAOR:** Blending the key observation features relevant to current hidden states via an FFN-like key-value retrieval.
>
> *   **Trigger Policies:**
>     *   **All Layers:** Injects observation features at every layer of the LLM backbone.
>     *   **Random:** Selects a subset of layers uniformly at random for each inference step. To ensure a fair comparison, the number of selected layers matches the average number of layers triggered by the *Entropy-based* policy (e.g., approximately **30%** for LIBERO-Spatial, Object, and Goal, and **20%** for LIBERO-Long).
>     *   **Entropy-based:** Dynamically triggers injection only at specific layers where the uncertainty measured by action entropy exceeds the threshold $\gamma$, targeting moments of high uncertainty.
>
> **(2) Quantitative Results**
>
> **Table 5: Ablation Study on Injection Mechanisms and Trigger Policies**
>
> |Method / Variant|Feature Extraction|Trigger Policy|Spatial|Object|Goal|Long|Avg.|
> |:-|:-:|:-:|:-:|:-:|:-:|:-:|:-:|
> |**OpenVLA-OFT**|-|-|98.2|98.2|97.6|94.2|97.1|
> |**Mean-Residual**|Mean Pooling|All Layers|0.0|0.0|0.0|0.0|0.0|
> |**Mean-Residual**|Mean Pooling|Random|98.0|98.4|96.8|94.4|96.9|
> |**Mean-Residual**|Mean Pooling|Entropy-based|0.0|0.0|0.0|0.0|0.0|
> |**Mean-Blending**|Mean Pooling|All Layers|98.0|96.8|95.8|94.4|96.3|
> |**Mean-Blending**|Mean Pooling|Random|98.4|97.8|97.8|94.8|97.2|
> |**Mean-Blending**|Mean Pooling|Entropy-based|98.0|97.8|97.6|93.8|96.8|
> |**UAOR (All Layers)**|Attentive Retrieval|All Layers|97.8|97.6|96.2|95.2|96.7|
> |**UAOR (Random)**|Attentive Retrieval|Random|97.8|97.6|96.4|93.6|96.4|
> |**UAOR (Ours)**|**Attentive Retrieval**|**Entropy-based**|**99.0**|**98.4**|**98.2**|**96.2**|**98.0**|
>
> **(3) Analysis and Discussion**
>
> * **Injection Mechanism (Collapse of Direct Addition):**
>
>   *   As shown in *Mean-Residual (All Layers/Entropy-based)*, direct addition leads to **catastrophic collapse (0.0%)**. This is because adding raw observation features directly to deep semantic hidden states significantly shifts the feature distribution. When applied consecutively (All Layers) or clustered (Entropy-based), the model cannot recover.
>   *   Interestingly, *Mean-Residual (Random)* survives (96.9%) because random injection is likely **sparse and non-consecutive**, allowing the model to recover from the disturbance in subsequent layers.
>   *   **Mean-Blending** is safer and prevents collapse, but its performance (Avg 96.3%-97.2%) still fails to surpass the strong baseline.
>
> * **Feature Extraction (Attentive vs. Mean):**
>
>   *   Even with safe blending, Mean Pooling variants generally underperform. This is because Mean Pooling assigns equal weight to all observation tokens, failing to distinguish relevant cues.
>   *   In contrast, **Attentive Retrieval** (UAOR) effectively extracts context relevant to the current hidden states, preserving fine-grained spatial cues essential for manipulation.
>
> * **Trigger Policy (Injecting when necessary):**
>
>   *   Given the high baseline (97.1%), **indiscriminate injection** introduces unnecessary perturbations to the hidden states. Both *UAOR (All Layers)* and *UAOR (Random)* degrade performance (96.7% / 96.4%) compared to the baseline.
>   *   **UAOR (Entropy-based)** is the **only** variant that achieves a gain (98.0%). It successfully targets the specific layers where observation information fades. By injecting only when necessary, it provides a targeted correction without interfering when the model is confident.
>
>   These findings collectively validate the effectiveness of the core designs of UAOR.

---

### Author Response · Authors · 2025-11-26
**General Response 1**

We sincerely thank all reviewers for their insightful comments and constructive feedback.  We are encouraged that they consistently recognized the contributions of our work: the **significance of the problem we studied** and the **compelling intuition** regarding observation decay leading to increased uncertainty in deep VLA layers (Reviewer surj, 7dgg, A2fr, rnbV); the simple yet efficient'' design of UAOR (Reviewer A2fr, rnbV);the solid and convincing theoretical analysis (Reviewer A2fr, rnbV); and the **systematic empirical validation** across heterogeneous VLA architectures, scales, and benchmarks (Reviewer 7dgg, A2fr, rnbV). We also appreciate the positive remarks on the **clarity and high writing quality** of our paper(R. surj, rnbV).

To thoroughly address the questions raised and further strengthen our paper, we have conducted extensive supplementary experiments totaling over 1,500 GPU hours, as we prioritized gathering comprehensive empirical evidence to ensure the utmost rigor. All major revisions in the updated manuscript are highlighted in blue. We hope these additional efforts and clarifications can help resolve the reviewers' concerns. Below is a summary of the major updates included in the revision:

### **1. Empirical Validation of Core Premise (New Fig. 2 & Revised Sec. 1)**
To address concerns regarding the link between uncertainty and observation forgetting (**Concern 1**), we added a **Layer-wise Attention Analysis** **(Fig. 2 & Sec. 1)**. Results show that the rise in Action Entropy perfectly correlates with the sharp decay of attention towards observation tokens, providing direct physical evidence for our hypothesis.

### **2. Rigorous Verification of the Rationale behind Action Entropy (New App. C.2 & Revised Sec. 4.3 & New Table 4)**
We validated the rationale of our uncertainty metric **Action Entropy** (**Concern 2**) through **Layer-wise Linear Probing** and comparisons against alternative metrics (Feature Entropy, Learned Heads) **(App. C.2)**. Results confirm that intermediate layers encode significant action semantics and that our training-free entropy metric serves as a reliable proxy for model confusion. Additionally, the new ablation on the trigger policy **(Sec. 4.3 & Table 4)** further proves that the Action Entropy trigger is essential.

### **3. Expanded Generalizability on SOTA Models (Revised Sec. 4.1 & Table 1 & Sec. 4.2 & Fig. 4)**
To demonstrate robustness beyond saturated baselines (**Concern 3**), we applied UAOR to the state-of-the-art flow-matching policy **$\pi_0$** (achieving **+1.5%** avg. gain) **(Sec. 4.1 & Table 1)** and conducted additional **real-world experiments on CogACT** **(Sec. 4.2 & Fig. 4)**, validating efficacy across diverse architectures and environments.

### **4. Comprehensive Design Ablations (Revised Sec. 4.3 & Table 4 & App. C.1)**
We conducted a **$2\times3$ factorial ablation study** **(Sec 4.3 & Table 4)** to vvalidate the effectiveness of UAOR's core designs, including a comparison against standard residual connections (Mean-Residual). Also we conducted a detailed ablation on **Injection Timing/Location** **(App. C.1)**, empirically verifying
the necessity and efficiency of injecting into the next layer’s FFN. These experiments rigorously justify our specific design choices, such as the "Next-Layer FFN" injection and the entropy-based trigger.

### **5. Theoretical, Methodological & Efficiency Refinements (Revised Sec. 3.3, 3.4, 4.4 & App. D)**
We clarified the sequential causality in our algorithmic formulation **(Sec. 3.3)** and unified the theoretical analysis under a cohesive **"Mechanism-Effect-Objective-Control"** framework **(Sec. 3.4)**. and extended our analysis of token-level weighting to include **Proprioception-guided schemes** **(Sec. 4.4)**, providing deeper insights into why uniform injection remains the most robust strategy. We also rigorously revised the **theoretical FLOPs complexity analysis** (**App. D**), quantitatively confirming the negligible computational overhead of UAOR.

For clarity and to avoid repetition, we begin by consolidating the following **major concerns** that were commonly raised across reviewers.

---

> ### Author Response · Authors · 2025-11-26
> **General Response 2**
>
> > ### **Concern 1: The core premise of the paper is not well-supported (The explanation of why forgetting leads to uncertainty lacks a clear reasoning process)**
>
> We thank the reviewers for this insightful critique. We agree that the relationship between "observation forgetting" and "action uncertainty" requires rigorous validation beyond simple intuition.
>
> To address this, we have incorporated **new empirical evidence** (Layer-wise Attention Analysis) and revisited our **theoretical foundations** to substantiate our core premise from three perspectives:
>
> **1. Empirical Evidence: Uncertainty Correlates with Attention Decay**
> To visualize the "forgetting" process, we conducted a **Layer-wise Attention Analysis** (as shown in **Figure 2** in the revised manuscript) on the OpenVLA-OFT baseline. We tracked the attention weights from **Action Tokens** to **Observation **, **Language and Action Tokens** across layers.
>
> *   **Observation:** We observed that attention to observation tokens **drops to a minimal level and remains low** (blue line) in the early-to-middle layers (Layers 2–8), while attention to language tokens remains dominant.
> *   **Correlation:** Crucially, this phase of "observation attention decay" perfectly aligns with the layers where **uncertainty (action entropy)** **rises and remains high** (as shown in Figure 1).
> *   **Conclusion:** This provides direct empirical evidence that the rise in uncertainty is not random noise but a symptom of the model losing its focus on observation cues ("forgetting") while struggling to reconcile conflicting language priors.
>
> **2. Logical Explanation: "Blind Confidence" vs. "Grounded Confidence"**
> Reviewers correctly noted that uncertainty eventually drops to near-zero in the final layers, which we think is an inherent result of the **LLM training paradigm**: the Cross-Entropy loss and the Softmax bottleneck force the probability distribution to collapse onto a deterministic token prediction. However, we argue that **how** this certainty is achieved matters. We think there are two distinct pathways:
>
> *   **Pathway 1 (Baseline - Ungrounded):** As visualized in our new **Figure 2**, the baseline model suffers from "observation forgetting" in the early layers (Action-to-Observation attention drops). To satisfy the requirement of the output layer to be "certain," the model compensates by over-relying on **Language Priors** (Action-to-Language attention remains high). For example, it might predict "close gripper" simply because that is the statistically likely next token in the language sequence, not because it visually confirms the object is grasped. This leads to **"blind confidence"**—the model is certain, but ungrounded, often resulting in hallucinations or execution failures.
> *   **Pathway 2 (UAOR - Grounded):** By reinjecting observation features when uncertainty spikes, UAOR forces the model to re-attend to the observation context. The uncertainty still drops at the final layer, but this certainty is now more **"grounded"** in the refreshed observation cues.
>
> **3. Theoretical Foundation: Information Gain guarantees Uncertainty Reduction**
> Beyond empirical results, our method is grounded in Information Theory (detailed in **Section 3.4**). We formally prove that injecting observation features mathematically reduces uncertainty:
>
> *   **Theorem 3.1 (Observation Information Gain):** We prove that UAOR increases the mutual information between the hidden state $\boldsymbol{h}$ and observation $\boldsymbol{o}$:
>     $$I(\hat{\boldsymbol{h}}_t^{(\ell+1)};\boldsymbol{o}_t) \ge I(\tilde{\boldsymbol{h}}_t^{(\ell+1)};\boldsymbol{o}_t)$$
>     This ensures that the reinjection effectively counteracts the "forgetting" or information loss.
> *   **Theorem 3.2 (Action Uncertainty Reduction):** We further show that increasing this mutual information strictly reduces the conditional entropy of the action:
>     $$H(\boldsymbol{y}_t \mid \hat{\boldsymbol{h}}_t^{(\ell+1)}) \le H(\boldsymbol{y}_t \mid \tilde{\boldsymbol{h}}_t^{(\ell+1)})$$
>     This theoretical result aligns with our empirical finding: by restoring observation information (Theorem 1), we naturally lower the uncertainty of the action distribution (Theorem 2).
>
> **Summary**
> Combining the **attention correlation (Figure 2)**, the **grounding pathway logic**, and the **theoretical proofs (Theorems 1 & 2)**, we believe the premise that "observation forgetting leads to uncertainty (and reinjection fixes it)" is now more robustly supported.

---

> ### Author Response · Authors · 2025-11-26
> **General Response 3**
>
> >### **Concern 2: The rationale and validity of Action Entropy**
>
> We thank the reviewers for their rigorous scrutiny. We clarify the rationale and validity behind **Action Entropy** from theoretical, empirical, and practical perspectives, supported by **new probing experiments** and **ablation studies**.
>
> **Note:** We apologize for the terminology error in the original manuscript referring to the projector as the "Last Layer's MLP." In practice, we use the **Language Modeling Head (LM Head)**, which is the **sole component** capable of projecting hidden states into the vocabulary space to derive probability distributions. We have corrected this error in our revised paper. Our design of applying the model's final lm head to decode intermediate hidden states is a standard practice in the "Logit Lens" paradigm [1], a widely validated and adopted approach in interpretability research.
>
> **1. Theoretical Validity: The Logic Lens View**
> Reviewer surj raised a fair concern by drawing an analogy to classifier guidance in diffusion models (where classifiers trained on clean images fail on noisy inputs). However, we respectfully submit that **Transformer dynamics differ fundamentally**:
>
> *   **In-Distribution Stream:** Transformers employ a **Residual Stream** architecture ($x_{l+1}=x_l+f(x_l)$) where representations are iteratively refined in a shared space. Unlike diffusion models, which suffer from a severe OOD mismatch (clean classifiers on noisy inputs), our intermediate states are generated by the same backbone under **standard inputs**. Thus, applying the LM head for "early decoding" aligns with established **linear probe** and **early exit** practices, rather than forcing a classifier to operate in a "noise domain." Therefore, we believe using their entropy as a relative difficulty indicator that does not need to be perfectly precise is reasonable.
> *   **Logit Lens Support:** Recent interpretability research (the "Logit Lens" [1] and "Tuned Lens" [2])  confirm that intermediate layers in Transformers often reside in a space aligned with the output, allowing for meaningful "early decoding". Furthermore, the feasibility of utilizing intermediate states is supported by recent works in the VLA domain, such as **DeeR-VLA** [3]. It employs dynamic early-exit mechanisms to decode actions directly from intermediate layers, confirming that intermediate representations preserve significant action-relevant information rather than being meaningless. Valle et al [4] also propose that using the entropy to measure the uncertainty of VLA models is a meaningful operation.
> *   **Relative Signal:** Crucially, intermediate predictions are used solely to derive a **scalar uncertainty score** for triggering reinjection, not for execution. Unlike diffusion guidance, we do not rely on **absolute correctness** or calibration. We use entropy as a **relative indicator** of "confusion," where high entropy signals that the model has not yet formed a consensus on the action.
> *   **Correlation Evidence:** Empirically, as shown in our new **Figure 2**, the **increase and sustained high level** of entropy in the early layers **align closely** with the **sharp drop and persistently low level** of observation attention. This suggests that the entropy is not merely random noise, but serves as a plausible indicator of the model's confusion arising from observation information dilution.
>
> **2. Empirical Verification: Layer-wise Probing**
>
> To further validate the rationale of computing action entropy from intermediate layers, we followed the reviewers's suggestion to **fine-tune linear lm heads (action head)** at intermediate layers (**added into the new Appendix C.2**).
>
> *   **Setup:** We froze the transformer backbone of OpenVLA-OFT and trained a separate linear action head for each selected layer (Layers 4, 8, ..., 32) to predict actions. The results are as follows:
>
> **Table 1: Layer-wise Action Prediction Success Rate with the fine-tuned lm head (OpenVLA-OFT on LIBERO-Long)**
>
> |Layer Index|4|8|12|16|20|24|28|32|
> |:-:|:-:|:-:|:-:|:-:|:-:|:-:|:-:|:-:|
> |**Success Rate**|45.6|62.3|78.5|86.1|90.4|92.1|93.5|94.2|
>
> *   **Analysis:**
>     The **considerable**  success rate in early-to-mid layers  confirms that intermediate hidden states already contain significant action-relevant information. This validates our design of using the frozen LM head as a **"rough decoder"**: since the features are semantically aligned with the action space, the resulting entropy serves as a reliable proxy for the model's current uncertainty.
>
> [1] Nostalgebraist. "Interpreting GPT: the logit lens." 2020.
>
> [2] Belrose et al. "Eliciting Latent Predictions from Transformers with the Tuned Lens." *arXiv*, 2023.
>
> [3] Yue et al. "DeeR-VLA: Dynamic Inference of Multimodal Large Language Models for Efficient Robot Execution." *NeurIPS*, 2024.
>
> [4] Valle et al. "Evaluating Uncertainty and Quality of Visual Language Action-enabled Robots". *arXiv*, 2025.

---

> ### Author Response · Authors · 2025-11-26
> **General Response 4**
>
> > ### **Concern 2: The rationale and validity of Action Entropy**
>
> **3. Design Choice: Metric Comparison**
>
> Reviewers have mentioned why we project to the action space rather than using **Feature Entropy** or a **Learned Uncertainty Head**. To address this, we implemented these variants and compared them on the LIBERO benchmark (**added as a new ablation study into section 4.3 of the revised paper**).
>
> *   **Feature Entropy:** Computes the entropy of the hidden state vector itself (after Softmax normalization).
> *   **Learned Uncertainty Head:** Trains a lightweight Linear layer (lm head) on collected hidden states to predict the binary correctness of the action (Offline Probing).
> *   **Action Entropy (Ours):** Uses the frozen LM head to project to the vocabulary space.
>
> **Table 2: Comparison of Uncertainty Metrics**
>
> |Uncertainty Metric|Training|Spatial|Object|Goal|Long|Avg.|
> |:-|:-:|:-:|:-:|:-:|:-:|:-:|
> |OpenVLA-OFT (Base)|-|98.2|98.2|97.6|94.2|97.1|
> |Feature Entropy|No|97.8|97.6|97.8|94.2|96.9|
> |Learned Head|Yes|98.8|98.2|98.2|95.8|97.7|
> |**Action Entropy (Ours)**|**No**|**99.0**|**98.4**|**98.2**|**96.2**|**98.0**|
>
> **Analysis:**
>
> *   **Why Feature Entropy is Ineffective (96.9% $\approx$ Base):** Hidden states are high-dimensional semantic embeddings, not probability distributions. Our layer-wise analysis reveals a **counter-intuitive trend** for Feature Entropy: it remains negligible ($\approx 0.0$) in middle layers but **spikes drastically** in the final layers (e.g., jumping from $0.04$ at Layer 28 to $0.79$ at Layer 32).
>
>     **Table 3: Layer-wise Feature Entropy (OpenVLA-OFT on LIBERO-Long)**
>
>     |Layer|1|2|3|4|5|6|7|8|9|10|11|12|13|14|15|16|
>     |:-|:-:|:-:|:-:|:-:|:-:|:-:|:-:|:-:|:-:|:-:|:-:|:-:|:-:|:-:|:-:|:-:|
>     |**Entropy**|0.000|0.066|0.086|0.094|0.105|0.097|0.052|0.012|0.008|0.002|0.000|0.001|0.001|0.001|0.000|0.000|
>
>     |Layer|17|18|19|20|21|22|23|24|25|26|27|28|29|30|31|32|
>     |:-|:-:|:-:|:-:|:-:|:-:|:-:|:-:|:-:|:-:|:-:|:-:|:-:|:-:|:-:|:-:|:-:|
>     |**Entropy**|0.000|0.000|0.000|0.000|0.000|0.000|0.006|0.010|0.009|0.014|0.026|0.036|0.042|0.174|0.577|0.789|
>
>     *   **Mismatch with Uncertainty:** True decision uncertainty should decrease as the model converges. The spike in Feature Entropy at deep layers represents **"richer feature activation"** (distributed representation) rather than confusion.
>     *   **Consequence:** Using this metric fails to trigger in the confusing middle layers while erroneously triggering in the final layers where the model is confident, acting as harmful noise that degrades performance. Due to the soft $\alpha$-blending, this late injection is not catastrophic but offers **no performance gain**, rendering the metric **ineffective**.
>
> *   **Learned Head vs. Action Entropy:**
>
>     * The **Learned Head** performs well (97.7%) but requires **additional training** and labeled data (collected hidden states with correct/incorrect actions), making it less practical.
>
>     * **Action Entropy (Ours)** outperforms it (**98.0%**) while being completely **training-free**. This is likely because the frozen LM head is already perfectly aligned with the model's pre-trained knowledge, offering a more intrinsic measure of the model's "confusion" than an auxiliary head trained on limited downstream data.
>
>     * **Efficiency:** UAOR uses the existing, frozen LM Head. Since we only project few or single **Action/Condition Tokens** (not the -whole sequence) at each layer, the computational overhead is **negligible**, making it highly efficient compared to training auxiliary heads.
>
>       We provide a theoretical FLOPs comparison based on OpenVLA-OFT (Llama-2-7B) and CogACT (Llama-2-7B) (**newly added into Appendix D**):
>       * **Notation:** Hidden dimension $D=4096$, Vocabulary size $D_v=32000$, Sequence length: OpenVLA-OFT ($N \approx 600$), CogACT ($N \approx 300$).
>
>       * **Action Tokens:** OpenVLA-OFT ($N_a=56$), CogACT ($N_a=1$, predicting a condition token per step).
>
>       *   **LM Backbone Cost:** Based on the estimation in Appendix D, the FLOPs for the language model backbone is:
>           $$C_{\text{LM}} \approx L\cdot (24ND^2 + 4N^2D)$$
>
>       *   **UAOR Projection Cost:** Projecting the action token hidden states to the vocabulary space (we don't project at the last layer since we need to reinject at the next year):
>           $$C_{\text{proj}} = (L-1) \cdot 2 \cdot N_a \cdot D \cdot D_v$$
>
>       * **Overhead Brought by Projection:**
>
>         $$r_{\text{proj}} \approx \frac{C_{\text{proj}}}{C_{\text{LM}}}$$
>
>       *   **OpenVLA-OFT**: $r_{\text{proj}} \approx 5.7\\%$
>
>           **CogACT:** $r_{\text{proj}} \approx 0.2\\%$
>
>       **Comparison:**
>
>       This confirms that the overhead introduced by the projection to compute action entropy is relatively low, which is negligible compared to the cost of training auxiliary heads or processing additional modalities.

---

> ### Author Response · Authors · 2025-11-26
> **General Response 5**
>
> > ### **Concern 2: The rationale and validity of Action Entropy**
>
> **4. Quantitative Verification: Trigger Policy Ablation**
>
> The ultimate test of a metric is its utility. We refer to our new **$2 \times 3$ Factorial Ablation Study** (**Table 4 in the revised manuscript**) to prove that the Action Entropy trigger is essential.
>
> * **Trigger Policies:**
>   *   **All Layers:** Injects observation features at every layer of the LLM backbone.
>   *   **Random:** Selects a subset of layers uniformly at random for each inference step. To ensure a fair comparison, the number of selected layers matches the average number of layers triggered by the *Entropy-based* policy (e.g., approximately **30%** for LIBERO-Spatial, Object, and Goal, and **20%** for LIBERO-Long).
>   *   **Entropy-based:** Dynamically triggers injection only at specific layers where the uncertainty measured by action entropy exceeds the threshold $\gamma$, targeting moments of high uncertainty.
>
> **Table 4: Ablation Study on Trigger Policy (subset)**
>
> | Method / Variant   |   Feature Extraction    |  Trigger Policy   | Spatial  |  Object  |   Goal   |   Long   |   Avg.   |
> | :----------------- | :---------------------: | :---------------: | :------: | :------: | :------: | :------: | :------: |
> | OpenVLA-OFT (Base) |            -            |         -         |   98.2   |   98.2   |   97.6   |   94.2   |   97.1   |
> | UAOR (All Layers)  |   Attentive Retrieval   |    All Layers     |   97.8   |   97.6   |   96.2   |   95.2   |   96.7   |
> | UAOR (Random)      |   Attentive Retrieval   |      Random       |   97.8   |   97.6   |   96.4   |   93.6   |   96.4   |
> | **UAOR (Ours)**    | **Attentive Retrieval** | **Entropy-based** | **99.0** | **98.4** | **98.2** | **96.2** | **98.0** |
>
> *   **Injecting when necessary:**
>     *   Given the high baseline (97.1%), **indiscriminate injection** introduces unnecessary perturbations to the hidden states. Both *UAOR (All Layers)* and *UAOR (Random)* degrade performance (96.7% / 96.4%) compared to the baseline.
>     *   **UAOR (Entropy-based)** is the **only** variant that achieves a gain (98.0%). It successfully targets the specific layers where observation information fades. By injecting only when necessary, it provides a targeted correction without interfering when the model is confident.
>
> * **Theoretical Support (Theorem 3.4 in Section 3.4)**
>   This is further supported by our theoretical analysis (Theorem 4 in Sec 3.4), which proves that If the entropy-based layer uncertainty $u_t^{(\ell)}$ correlates positively with $H(\boldsymbol{y}_t \mid \tilde{\boldsymbol{h}}_t^{(\ell+1)})$, then conditioning reinjection on $u_t^{(\ell)} > \gamma$ increases the expected relevance of injected information:
>   $$\mathbb{E}\left[I\left(INJ_t^{(\ell+1)};\boldsymbol{y}_t \mid \tilde{\boldsymbol{h}}_t^{(\ell+1)}\right) \big| u_t^{(\ell)} > \gamma\right]
>   \ \ge\
>   \mathbb{E}\left[I\left({INJ}_t^{(\ell+1)};\boldsymbol{y}_t \mid \tilde{\boldsymbol{h}}_t^{(\ell+1)}\right)\right]$$
>
>   This theorem guarantees that **selective injection** (triggered by entropy) yields higher information gain 	than indiscriminate injection.

---

> ### Author Response · Authors · 2025-11-26
> **General Response 6**
>
> > ### **Concern 3: On the magnitude of performance gains**
>
> We appreciate the reviewers' candid feedback regarding the magnitude of the performance gains. While the numerical improvements might appear modest at first glance, we respectfully argue that they are significant given the context of the strong baselines and the **efficiency** of our method.
>
> - **Saturated Baselines:** The models we compare against (e.g., OpenVLA-OFT, CogACT, LLaVA-VLA) are already strong baselines with high performance, leaving limited room for marginal improvement.
>
> - **Efficiency & Practicality:** The core contribution of UAOR is achieving these gains **without** any additional training, fine-tuning, or auxiliary observation inputs (e.g., depth, point clouds).
>
> - **Comparison with methods requiring finetuning:** To illustrate this advantage, we compare UAOR with a recent method, **3D-CAVLA** [1] (**added in Table 1 of the revised manuscript**), which integrates CoT reasoning, depth perception, and ROI detection. While 3D-CAVLA achieves an average success rate of **98.1%** on LIBERO based on OpenVLA-OFT, it requires computationally expensive depth processing, LLM CoT reasoning, and additional fine-tuning. In contrast, UAOR achieves a similar **98.0%** performance solely through inference-time intervention with negligible computational overhead. This highlights that our "modest" gain actually represents a significant leap in **efficiency-to-performance ratio**.
>
> - **More significant gains on another strong baseline $\pi_0$:** To directly address the request for validation on more recent models and demonstrate that our gains are robust, we applied UAOR to $\pi_0$ [2], a cutting-edge flow-matching VLA policy. We evaluate under three random seeds and report the mean success rate and standard deviation. The results are as follows (**also added in Table 1 of the revised manuscript**):
>
>   **Table 5: Performance gains on $\pi_0$ on the LIBERO Benchmark**
>
>   | Method / Variant                        |     Spatial      |      Object      |       Goal       |       Long       |     Average      |
>   | :-------------------------------------- | :--------------: | :--------------: | :--------------: | :--------------: | :--------------: |
>   | $\pi_0$ (Black et al., 2024) (*RSS'25*) |   96.3$\pm$0.6   |   96.7$\pm$0.7   |   92.9$\pm$1.2   |   80.5$\pm$1.2   |   91.7$\pm$0.5   |
>   | **w/ UAOR (Ours)**                      | **97.3**$\pm$0.2 | **98.5**$\pm$0.2 | **94.3**$\pm$0.2 | **82.5**$\pm$0.5 | **93.2**$\pm$0.1 |
>   | $\Delta$                                |       +1.0       |       +1.8       |       +1.4       |       +2.0       |       +1.5       |
>
>   *   **Analysis:** As shown in Table 5, UAOR consistently improves the strong $\pi_0$​ baseline across all task suites, achieving an average gain of **+1.5%**, with a notable **+2.0%** improvement on the challenging Long-horizon tasks. At the same time, the standard deviation remains low across all task suites, demonstrating that UAOR leads to more robust and reliable performance with minimal variance, even under random initialization.
>   *   **Conclusion:** This demonstrates that UAOR is not limited to specific architectures but serves as a general-purpose booster. On models that have not yet hit the "saturation ceiling" (like $\pi_0$ on LIBERO-Long), UAOR unlocks even more substantial potential.
>
> [1] Bhat et al. "3D CAVLA: Leveraging Depth and 3D Context to Generalize Vision Language Action Models for Unseen Tasks." *arXiv*, 2025.
>
> [2] Black et al. "$π_0$: A Vision-Language-Action Flow Model for General Robot Control." *RSS*, 2025.

---

### Note · Authors · 2026-02-11

I have read and agree with the venue's withdrawal policy on behalf of myself and my co-authors.

---

### Meta-Review · Area_Chair_tdV1 · 2026-01-07

**Summary:**

The paper has received mixed reviews. After going over the rebuttal and author responses, the AC feels that the paper remains in a borderline state. While some of the concerns from the reviewers were addressed satisfactorily, such as the modest performance gain on LIBERO, other concerns (which the AC feels are major) were not convincingly answered.

Remaining concerns include:
- Details about how the new Pi-0 baseline numbers were extracted are unclear. Those numbers are not reported in the original paper. These numbers are lower than reported in other papers like "Knowledge Insulating Vision-Language-Action Models: Train Fast, Run Fast, Generalize Better" from Physical Intelligence (Table 1).
- Similarly, the new real-world experiment on CogAct is less convincing as only 10 evals were used for reporting performance, which means that a 10% gap in performance in one task can come via just 1 difference in failure vs success. It might be helpful to have at least 20 evals or more per task so that the difference in performance is well established.

The idea seems nice, but the AC feels that the paper would gain a lot by addressing some of the remaining concerns. It is in a borderline state, and the authors are encouraged to revise and resubmit.

**Reviewer Concerns:**

Addressed:
- Modest performance gains.

Remaining:
- Limited real-world experiments, especially regarding the validity of the current setup and missing statistical confidence.
- Not applied to most recent SOTA models, with specific concerns around the Pi-0 numbers.

Unclear:
- Concerns around the validity of the core promise are not fully resolved. The AC sides with the reviewers that the justification is limited. The argument made by the authors in the rebuttal is unclear to the AC.

**Reviewer Scores:**

Reviewer surj - Likely unchanged
Reviewer 7dgg - Likely unchanged
Reviewer A2fr - Likely unchanged
Reviewer rnbV - Likely unchanged

---

### Decision · Program_Chairs · 2026-01-26

Reject